# An event map of memory space in the hippocampus

**Lorena Deuker[1,2]\*, Jacob LS Bellmund[1,3], Tobias Navarro Schröder[1,3], Christian F Doeller[1,3]\***

[1]Donders Institute for Brain, Cognition and Behaviour, Radboud University Nijmegen, Nijmegen, The Netherlands; [2]Department of Neuropsychology, Institute of Cognitive Neuroscience, Ruhr-University Bochum, Bochum, Germany; [3]Kavli Institute for Systems Neuroscience, Centre for Neural Computation, Egil and Pauline Braathen and Fred Kavli Centre for Cortical Microcircuits, NTNU - Norwegian University of Science and Technology, St. Olavs University Hospital, Trondheim, Norway

**Abstract** The hippocampus has long been implicated in both episodic and spatial memory, however these mnemonic functions have been traditionally investigated in separate research strands. Theoretical accounts and rodent data suggest a common mechanism for spatial and episodic memory in the hippocampus by providing an abstract and flexible representation of the external world. Here, we monitor the de novo formation of such a representation of space and time in humans using fMRI. After learning spatio-temporal trajectories in a large-scale virtual city, subject-specific neural similarity in the hippocampus scaled with the remembered proximity of events in space and time. Crucially, the structure of the entire spatio-temporal network was reflected in neural patterns. Our results provide evidence for a common coding mechanism underlying spatial and temporal aspects of episodic memory in the hippocampus and shed new light on its role in interleaving multiple episodes in a neural event map of memory space.

**\*For correspondence:** lorena. deuker@rub.de (LD); christian. doeller@donders.ru.nl (CFD)

**Competing interests:** The authors declare that no competing interests exist.

## Introduction

The hippocampus is one of the most extensively studied regions in the brain. However, two of its core functions, spatial navigation and episodic memory, have mostly been investigated in separate research lines (*Eichenbaum, 2014*). It has been suggested that the answer to the apparent duality in hippocampal function resides in a common mechanism that is required for both spatial navigation and episodic memory (*Eichenbaum, 2014*): the formation of an abstract representation of the external world, a memory space (*Eichenbaum et al., 1999*). While it is clear that such a map-like representation would be necessary for spatial navigation, it might be less obvious for episodic memory. Yet, episodic memory has been defined as the ability to recall events from one's own life (*Tulving, 1983*) in a specific mode of retrieval that has been referred to as recollection (*Eichenbaum et al., 2007*) or 'mental time travel' (*Tulving, 2002*). This specific mode of retrieval makes it necessary that humans can, in their minds, re-create and re-experience episodes of their past by mentally navigating to the point when and where the episode happened, thereby retrieving the time and the place of past events. Notably, this implies that humans must be able to convert relationships between events, for example along the physical dimensions of space and time, into a mental representation so that the arrangement of events is appropriately reflected. In line with this idea, recent discoveries in rodent electrophysiology indicate that cells in the hippocampus code for events in space and time simultaneously (*Kraus et al., 2013*, *2015*; *Mankin et al., 2012*) and provide evidence for the notion that memories are, in fact, stored in a multi-dimensional memory space

(*Eichenbaum et al., 1999*; *McKenzie et al., 2014*). Findings from fMRI studies in humans also suggest that memories are dynamically integrated into mnemonic network representations along different dimensions (*Collin et al., 2015*; *Horner et al., 2015*; *Kumaran and Maguire, 2006*; *Milivojevic et al., 2015*; *Preston and Eichenbaum, 2013*; *Schlichting et al., 2015*; *Shohamy and Wagner, 2008*; *Zeithamova et al., 2012*). However, it remains elusive how inter-event relationships along multiple dimensions, such as *space* and *time*, are combined and converted into a multi-dimensional mnemonic event map, which might potentially support episodic memory.

In studies with rodents, the hippocampus has traditionally been associated with the representation of space (*Burgess et al., 2002*; *Moser et al., 2008*; *O'Keefe and Nadel, 1978*), i.e. knowing where events occurred. This spatial code is supported by specific neurons in the hippocampal formation such as place cells (*O'Keefe and Dostrovsky, 1971*), which increase firing rate when a specific place in an environment is traversed, and grid cells, which increase firing rate in multiple locations, organized in a hexagonal grid pattern (*Hafting et al., 2005*). Although much of this research has employed electrophysiology in rodents, recent studies using functional magnetic resonance imaging have pointed to similar mechanisms in humans (*Doeller et al., 2010*; *Hassabis et al., 2009*; *Howard et al., 2014*; *Kyle et al., 2015*; *Vass and Epstein, 2013*; *Wolbers et al., 2007*).

In addition to its role in spatial representation, the hippocampus is also known to be crucial for episodic memory in humans (*Eichenbaum, 2014*; *Norman and O'Reilly, 2003*; *Scoville and Milner, 1957*; *Stark and Squire, 2000*). While it has been acknowledged that episodic memory is inherently structured in a temporal manner (*Tulving, 1985*), our understanding of the role of the hippocampus for remembering temporal structure has been limited for a long time (*Howard and Eichenbaum, 2015*). Recently, descriptions of situation-specific cell-assembly firing sequences (*Pastalkova et al., 2008*) and the discovery of time cells in the rodent hippocampus (*MacDonald et al., 2011*) have sparked a renewed interest in the role of the hippocampus in temporal memory. Time cells have been shown to selectively increase their firing pattern at specific time points after a cue had been presented, while animals were waiting to respond to an odor (*MacDonald et al., 2011*). Interestingly, the cells exhibited this temporally coordinated firing pattern only in specific contexts. In a new context, e.g. when delay time was prolonged, some of the previously responding cells ceased to fire or fired at different time points while previously unresponsive cells suddenly began to code for elapsed time, a response pattern akin to spatial remapping of place cells ('retiming'). Thus, rather than displaying a simple counting function or a generic delay signal, these cells seem to represent the specific temporal context of an episode, consistent with the temporal context model (*Howard and Kahana, 2002*; *Howard et al., 2005*). These findings have led to a re-examination of the hippocampus' role in temporal memory in rodents and humans (*Eichenbaum, 2014*; *DuBrow and Davachi, 2015*; *Ranganath and Hsieh, 2016*) and to several recent neuroimaging studies in humans. For example, one study found that, in predictable as opposed to random sequences of items, items that are closer together in the sequence elicit increased neural pattern similarity (*Hsieh et al., 2014*), an effect which is dependent on the conjunction of item identity and order position rather than order position alone. Another study showed that items are represented differently within event boundaries than across event boundaries (*Ezzyat and Davachi, 2014*). Interestingly, participants' judgment of temporal distance between a pair of items was systematically higher when the pair was separated by an event boundary compared to when it was within an event boundary (even though the actual temporal distance was the same for the two types of item pairs). Further, a subsequent behavioral judgment of across-boundary item pairs as being temporally close was associated with higher pattern similarity during the learning task when compared to across-boundary items that were judged to be far. However, the temporal sequences in these studies were investigated independently of the spatial relationship between the elements. Another recent report showed that spatial and temporal aspects of autobiographical experiences are coded within the hippocampus across various scales of magnitude, up to one month in time and 30 km in space (*Nielson et al., 2015*). Participants wore a camera over the course of four weeks, which automatically took pictures throughout the day. Later, participants were scanned with fMRI while reviewing these pictures and trying to recall what they depicted. While providing interesting insight into real-life autobiographical memory, there was little experimental control over the stimulus material with regards to visual properties and the degree of familiarity participants had with the locations. More importantly, none of the studies mentioned above compared changes in neural pattern similarity from before the acquisition of the spatial and temporal structure to after.

The present study tests the overarching idea that the hippocampus represents events within a multi-dimensional event map of memory space (*Eichenbaum et al., 1999*). More specifically, we will use the term 'event map' to refer to the mental representation of a complete set of interrelationships between events. In order to support an efficient and adaptive memory system, such an event map should be abstract, flexible and relational (*Eichenbaum, 2014*; *Eichenbaum et al., 1999*): Abstract in the sense that it represents aspects of the experience that go beyond a direct record of events, such as being able to extrapolate that taking a specific turn in a city will be a shortcut even before the actual experience has been made. An event map should further be flexible to allow for the representation of sudden changes in the world, such as roadblocks. Thirdly, it should be able to represent relationships along different dimensions concurrently and conjunctively, while still allowing to focus on one dimension depending on task situations, such as for example knowing that the spatial distance between two bus stops is short, but that it could take a long time to get to the destination during rush hour.

The goal of this study was to investigate whether experiencing multiple events within a spatio-temporal structure leads to the acquisition of a neural event map that is *abstract, flexible* and *relational*. We used a highly realistic first-person virtual navigation paradigm that led participants through a complex virtual city ('Donderstown', see http://www.doellerlab.com/donderstown/). The purpose of this task was to provide a learning experience for participants in which 16 objects were arranged consistently in a spatial and temporal structure, defined through the complex network of inter-object *relations*. We dissociated the dimensions of time and space through the use of teleporters, requiring a high level of *flexibility* in memory (see *Figure 1A* for details of the task). To ascertain maximal experimental control, the objects were shown repeatedly in random order before and after the learning task and all fMRI analyses were performed on data acquired during these independent scanning sessions. Knowledge of the spatio-temporal structure acquired during the learning task was assessed with a memory test after fMRI acquisition (see *Figure 1B*). Participants' *abstract* representation of the event structure was estimated through all possible pairwise spatial and temporal distance judgments, including judgments which required higher-order inference because events were separated by multiple intervening events. The participant-specific spatial and temporal distance ratings from the memory test were then used to investigate changes in neural pattern similarity from before to after the learning task leveraging representational similarity analysis (RSA; *Kriegeskorte et al., 2008*). More specifically, we investigated whether increases in hippocampal pattern similarity co-varied with the remembered spatial and temporal event structure during the learning task.

## Results

### Behavioral results

Participants had to learn the temporal and spatial relationships between 16 objects placed in boxes along a route (see *Figure 1* and *Figure 1—figure supplement 1* for details) by repeatedly navigating along the specific route in the virtual city environment Donderstown. In order to ensure sufficient learning of the spatio-temporal trajectory, participants had to complete 14 rounds of the route. Proficiency in this virtual-navigation task (see *Figure 1A*) was assessed by analyzing improvements during the task as well as by investigating performance in the subsequent memory tests. Both the spatio-temporal learning task and the subsequent memory tests were done outside of the MR scanner, so neural activity during these tasks cannot be assessed.

#### Spatio-temporal learning task

Participants required on average 71.63 ± 13.75 (mean ± std) minutes for the task (range 52.67–113.06 min), showing high variability in navigation speed. When looking at the time it took participants to get from one object to the next (*Figure 1—figure supplement 2*) across the 14 route repetitions, we observed a rapid decrease of navigation times over the first 3–4 repetitions, with navigation duration roughly converging to the time it takes to walk from one box to the next (raw walking time). Navigation times during the last repetition were significantly shorter than during the first repetition (21.55 vs 11.60 min, $T_{25}$ = 6.70, p<0.0001), see *Figure 1—figure supplement 2* for more details. In sum, these data indicate that participants were able to learn the virtual route.

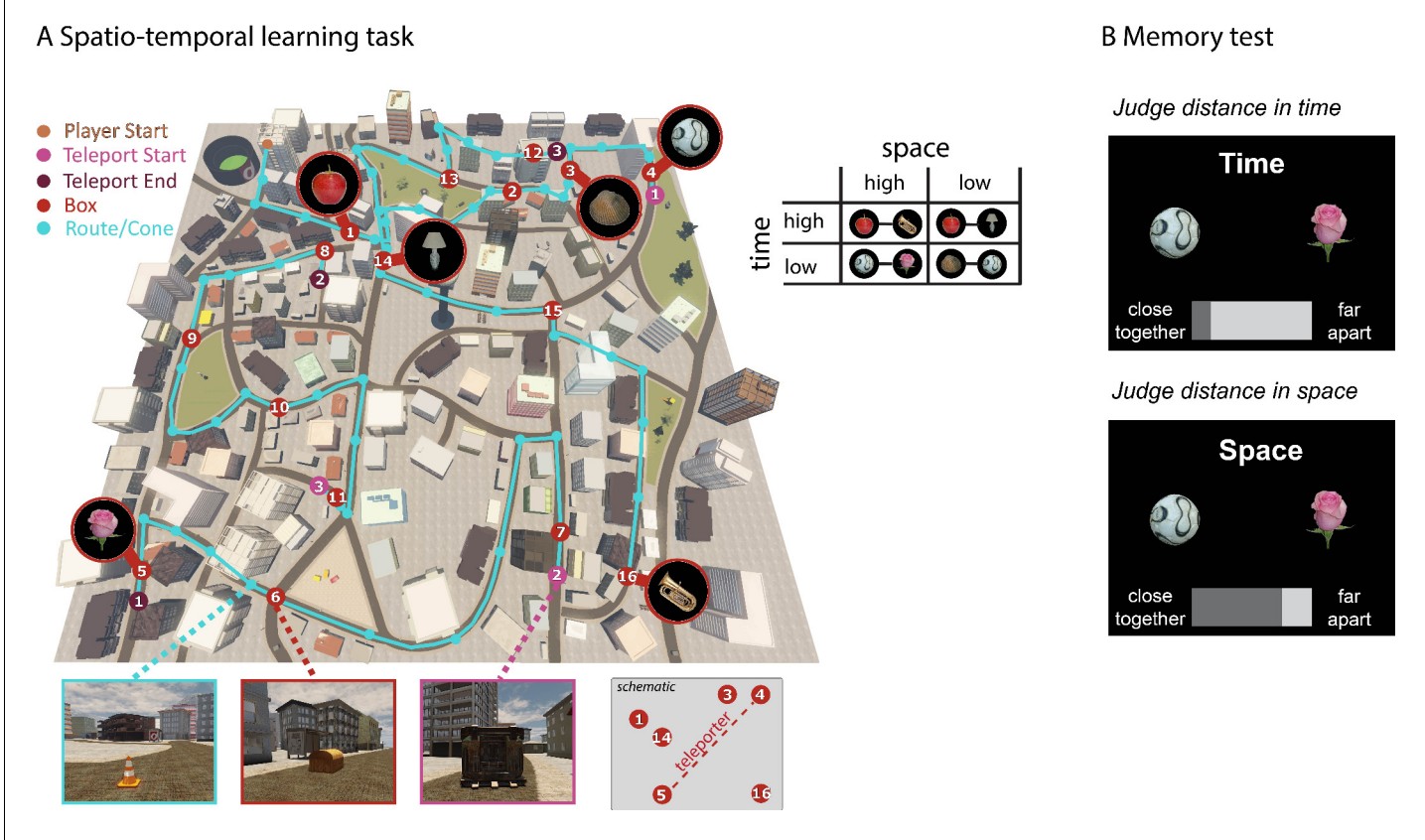

**Figure 1.** Learning spatio-temporal trajectories in virtual reality. (**A**) Overview of the route participants had to take through the virtual reality city Donderstown. 16 objects were presented along the route (see *Figure 1—figure supplement 1* for details on the objects). Participants were first guided by the presentation of traffic cones (marked here with turquoise circles) that led them from one wooden box (red numbered circles) to the next. The cones disappeared after 6 repetitions of the route (see *Figure 1—figure supplement 2* for behavioral performance in the navigation task). Crucially, the spatial and temporal distance between objects was systematically manipulated (see Materials and methods for details). As exemplified in the table, pairs of objects have either high or low spatial distance to one another as well as high or low temporal distance. At three points along the route, participants had to use a teleporter (pink and purple numbered circles), which transported them immediately from one part of the city to a completely different part of the city. Introducing the teleporters allowed us to have pairs of objects with a high spatial distance and low temporal distance, as can also be seen in *Figure 1—figure supplement 3*. (**B**) In a subsequent memory test outside the scanner, participants were asked to judge for every possible pair of objects how close together or far apart the objects had been in space (Euclidean distance) or time (how long it took them to get from one object to the next).

The following figure supplements are available for figure 1:

**Figure supplement 1.** Overview of the 16 objects used in the picture viewing tasks and the learning task.

**Figure supplement 2.** Performance during the VR learning task.

**Figure supplement 3.** Pairwise spatial and temporal distances are independent from each other.

## Memory tests

We assessed participants' memory of the spatio-temporal structure of objects with three different memory tests after MRI acquisition. Firstly, participants were asked to freely recall all objects they encountered during the task. Secondly, they had to indicate the spatial and temporal distance between every possible pair of objects. Thirdly, participants were given a schematic map of Donderstown on the computer screen, shown the image of every object and asked to indicate the location of the box that contained this object by moving the mouse, see Materials and methods for more details on the memory tasks.

## Free recall test

Participants recalled on average 13.08 ± 3.06 (mean ± std) of the 16 items. The order of free recall was more influenced by the temporal order during the task than by the spatial arrangement, as assessed by correlating the spatial and temporal distances between items during the task with the distance in order during free recall (mean R for spatial distance: −0.01 ± 0.16 [mean ± std]; temporal distance: 0.50 ± 0.40 [mean ± std]; $T_{25}$ = 4.32, p<0.0003).

## Distance judgment task

In this memory test, we asked participants to judge for every possible pair of objects how close together or far apart they had been presented during the learning task, both in space and in time. This yielded a participant-specific distance estimate for both the spatial and the temporal domain, effectively probing the participant's mnemonic event map. Crucially, by asking participants to make distance judgments for every possible pair of items, we required them to infer the spatial and temporal distances of items that had not been directly experienced together in the task. For every participant, we compared the subjective distance judgments with the objective distance during the task. Because memory distance judgments were given on a scale from 'close together' to 'far apart' rather than in absolute terms (see *Figure 1B*), accuracy was tested by the goodness of fit between the actual spatial and temporal distances in the learning task and the estimated spatial and temporal distances. For temporal judgments, memory distances were significantly correlated with actual temporal distances in 24 of the 26 participants (p<0.05; R = 0.64 ± 0.29 (mean ± std), see *Figure 2A* for the correlation coefficients across participants and *Figure 2—figure supplement 1* for participant-specific scatter plots). For spatial judgments, memory distances were significantly correlated with actual spatial distances in 21 of the 26 participants (p<0.05; mean ± std: R = 0.49 ± 0.29); *Figure 2A*). Thus, correspondence between actual and reproduced distances was very high for both space and time and slightly better for the temporal than for the spatial condition ($T_{25}$ = −2.52, p = 0.019). We also examined the relationship between participants' spatial and temporal distance judgments for a given pair of objects (even though the two factors were independent in the task, see Materials and methods). Indeed, we found that in 14 of the 26 participants, there was a significant correlation between their spatial and temporal distance judgments (p<0.05, R = 0.31 ± 0.29 [mean ± std]). Therefore, the two factors were not independent in most participants' memory judgments and we addressed this in our fMRI analysis (see below).

To further investigate the relationship between spatial and temporal distance judgments, we set up two GLMs which model the impact of actual spatial and actual temporal distances on (a) spatial distance ratings and (b) temporal distance ratings, respectively (see *Figure 2B*). Across participants, both actual spatial distances and actual temporal distances explained variance in spatial distance ratings (beta for the factor actual spatial distance: 0.50 ± 0.28 [mean ± std], significantly different from zero across participants: $T_{25}$ = 9.08, p<0.0001; beta for the factor actual temporal distance 0.17 ± 0.19 [mean ± std]; significantly different from zero across participants: $T_{25}$ = 4.48, p<0.001). However, the factor actual spatial distance had a significantly greater impact on spatial distance ratings than the factor actual temporal distance (t-test between betas across participants for space > time: $T_{25}$ = 4.06, p<0.001). Similarly, both actual spatial distances and actual temporal distances explained variance in temporal distance ratings (beta for the factor actual spatial distance: 0.16 ± 0.22 [mean ± std], significantly different from zero across participants: $T_{25}$ = 3.58, p<0.01; beta for the factor actual temporal distance 0.65 ± 0.27 [mean ± std]; significantly different from zero across participants: $T_{25}$ = 12.16, p<0.0001). The factor actual temporal distance had a much bigger impact on temporal distance ratings than the factor actual spatial distance (t-test between betas across participants for time > space $T_{25}$ = 5.33, p<0.0001). Thus, while there was some 'cross-over' between the domain that should be rated and the respective other domain, the domain that should be rated had a greater impact on the judgments both for space and for time.

We also investigated whether errors in judging spatial and temporal distances (i.e. the difference between z-scored actual distance and z-scored remembered distance) were systematically related to the distance in the other dimension (see *Figure 2C*). Indeed, we found that errors in spatial distance ratings were correlated both with actual temporal distance and remembered temporal distances (Fisher z-transformed correlation coefficients tested against zero across participants; actual temporal distance: $T_{25}$ = −5.39, p<0.0001; remembered temporal distance: $T_{25}$ = −5.33, p<0.0001). Similarly,

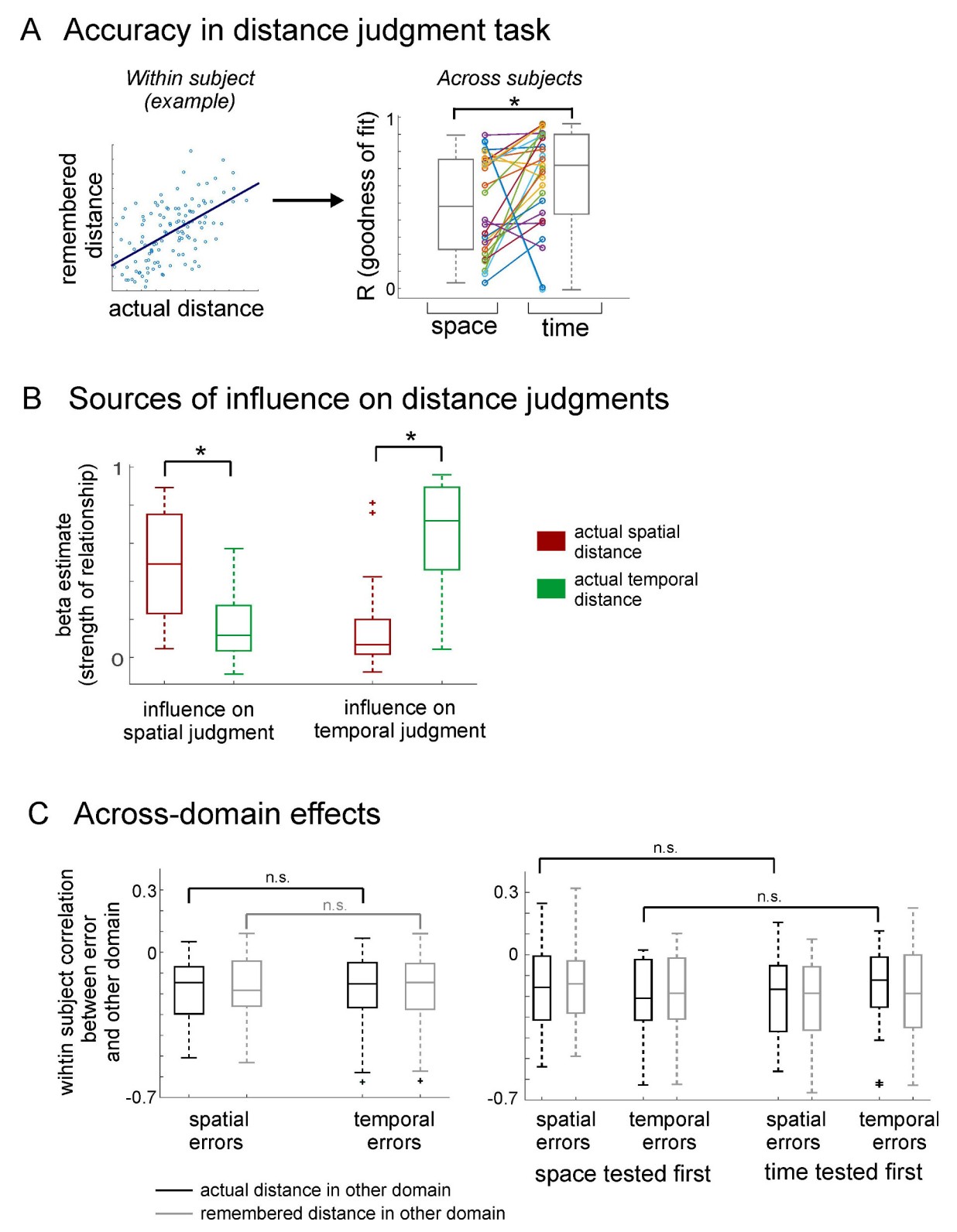

**Figure 2.** Results from the distance judgment task. (**A**) Accuracy in the distance judgment task was assessed by correlating the actual distance between pairs of items with the distance ratings given by participants during the memory task (illustrated for one participant as an example in the left panel; scatter plots for all participants can be found in *Figure 2—figure supplement 1*). The higher the correlation coefficient, the better the memory performance. Correlation coefficients for all participants are shown in a boxplot on the right side, both for the spatial and the temporal domain.
*Figure 2 continued on next page*

*Figure 2 continued*

Correlation coefficients are significantly different from zero across participants. Memory judgment for time was slightly better than for space. Individual participants' values are shown between the two boxplots, with lines connecting the corresponding values of the same participant. See *Figure 2—figure supplement 2* for exemplary results from a map test on spatial memory. (B) Result of two GLMs, modeling the impact of actual space and actual time on spatial distance ratings and temporal distance ratings, respectively. The boxplots show the beta estimates for the two factors across participants. Spatial judgments are related to actual distance in both space and time, and the same is true for temporal judgments. However, spatial distance has a higher impact than temporal distance on spatial judgments and temporal distance has a higher impact than spatial distance on temporal judgments. (C) Left: Investigating whether one domain biased the errors committed in the other domain, we correlated the errors in distance ratings with the actual or remembered distance in the other domain. Both time and space were correlated with errors committed in the other domain, but neither more strongly than the other. (C) Right: The same analysis as on the left side, but with trials split up depending on whether memory for space or time was tested first. The order in which memory was tested had no impact on the bias one domain had on errors committed in the other domain.

The following figure supplements are available for figure 2:

**Figure supplement 1.** Participants acquire knowledge about the temporal and spatial structure of events.

**Figure supplement 2.** Results from the subsequent map test.

errors in temporal distance ratings were correlated with both actual spatial distances and remembered spatial distances (Fisher z-transformed correlation coefficients tested against zero across participants; actual spatial distance: $T_{25} = -4.82$, p<0.0001; remembered spatial distance: $T_{25} = -5.04$, p<0.0001). It is conceivable that the bias that the opposite domain has on the distance rating of the domain that should be judged depends on which domain is tested first, e.g. it could be hypothesized that actual spatial distance has a higher impact on errors in temporal distance ratings when spatial distance was probed first within a participant. Therefore, we repeated the error analysis described before, but this time split trials up depending on whether space was tested first or time was tested first for a given pair of items, and then correlated the errors with the distance in the other domain separately. Neither the spatial domain nor the temporal domain was differentially affected by a bias from the other domain (neither actual nor remembered) depending on test order (all p>0.09). This means that the order in which spatial or temporal distances were probed did not have an impact on biases in the distance judgments.Taken together, these behavioral results show that participants were mostly accurate in reproducing the spatial and temporal structure between objects that had not been directly experienced together, indicating that they successfully formed an abstract, relational event map. While spatial and temporal distance ratings were correlated with one another in some participants, there is no evidence that either the spatial domain or the temporal domain had a bigger impact on distance ratings than the other, and that the degree to which one domain was biased by the other did not depend on which domain was tested first.

## Map test

On average, participants positioned the items with a distance error (expressed here as the ratio between the displacement error and the side length of the city map) of 0.193 ± 0.16 to the actual item location and performance was variable across participants (range 0.017–0.480, see *Figure 2—figure supplement 2*).

## Neuroimaging results

To assess the representational change as a consequence of the de novo acquisition of the spatial and temporal structure of events during the learning task, two independent picture-viewing tasks (PVT) inside the fMRI scanner preceded (PVT pre) and followed (PVT post) the spatio-temporal learning task (see *Figure 3* for an outline of the experimental sessions). In these two PVT fMRI blocks, participants saw pictures of the same 16 objects that were also presented during the spatio-temporal learning task (see *Figure 1—figure supplement 1*). Objects were presented multiple times on a black background, in random order, and participants were asked to press a button whenever they saw a target object (see Materials and methods).

The rationale behind the picture-viewing task was to assess the neural pattern similarity between pairs of objects without possible confounds of stimulus presentation during the learning task. If we

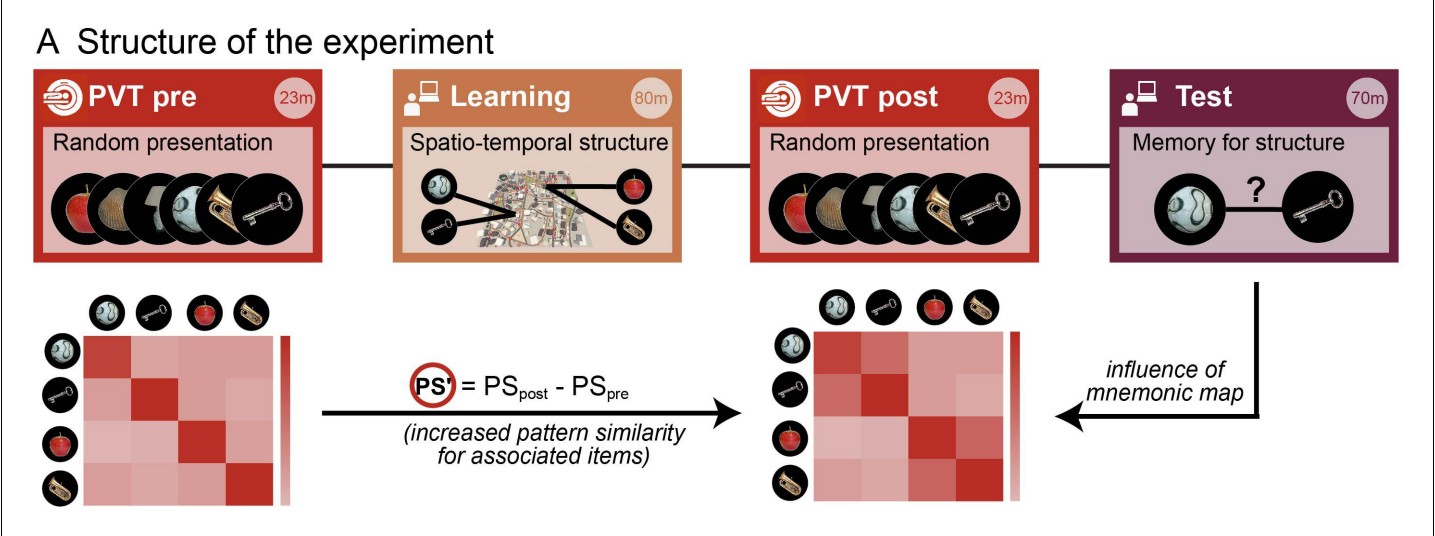

**Figure 3.** Assessing memory-related changes in neural similarity as a result of learning the spatio-temporal event structure. FMRI data were acquired during two blocks of an identical picture viewing task ('PVT', in red) before and after the virtual navigation learning task (gold). This allowed us to measure the fine-grained neural similarity structure between event representations. Event memories were subsequently assessed in separate memory tests for space and time (purple). The crucial index for assessing the spatial and temporal event structure as a result of the learning task was the change in neural similarity from before the learning task to after (expressed as PS') and how it covaried with the remembered spatial and temporal distances in the subsequent memory task.

had assessed pattern similarity for objects while they were presented in Donderstown during the learning task, analyses would have been susceptible to visual confounds for spatially close items (sharing more similar views of the environment) and possible auto-correlation confounds for temporally close items due to the slow hemodynamic response (i.e. temporally closer volumes will always be more similar to one another) or effects related to head movements. Analyzing pattern similarity in this independent task, when the objects were shown out of context and in random order, gave us high experimental control. Furthermore, both sessions, PVT pre and PVT post, were identical with respect to stimulus order and timing. Any changes in pattern similarity from PVT pre to PVT post (PS') are thus due to a changed neural representation of objects as a result of the spatio-temporal learning task and the newly formed memories. Therefore, we related the difference in pattern similarity from PVT pre to PVT post (PS') to the remembered temporal and spatial distances, both in a region of interest (ROI) analysis and a searchlight analysis (see *Figure 4* and Materials and methods for details on analyses and nonparametric statistical procedures). We pursued these approaches in parallel because they offer complementary advantages: the ROI approach allows for rigorous testing of a clear a priori hypothesis, while the searchlight approach allows us to identify possible regions outside of hippocampus that show the same effect, as well as to pinpoint any effect more locally within hippocampus.

## The hippocampus represents a spatio-temporal event map

We hypothesized that the hippocampus would support spatial and temporal event memory and, importantly, the combination of both. Therefore, in a first step, we investigated whether the change in pattern similarity across all hippocampal grey-matter voxels was related to participants' spatial and temporal distance judgments for pairs of items (see Materials and methods for details). We found that pattern similarity changes in bilateral hippocampus reflected participants' spatial distance judgments (Z = −3.719, $p_{FDR}$=0.0005; FDR correction for 15 multiple comparisons, see Methods and materials), as well as their temporal distance judgments (Z = −2.597, $p_{FDR}$ = 0.0078), see *Figure 5*. More specifically, pairs of objects which were recalled as being close together either in space or time during the task had higher pattern similarity increases across all hippocampal grey-matter voxels.

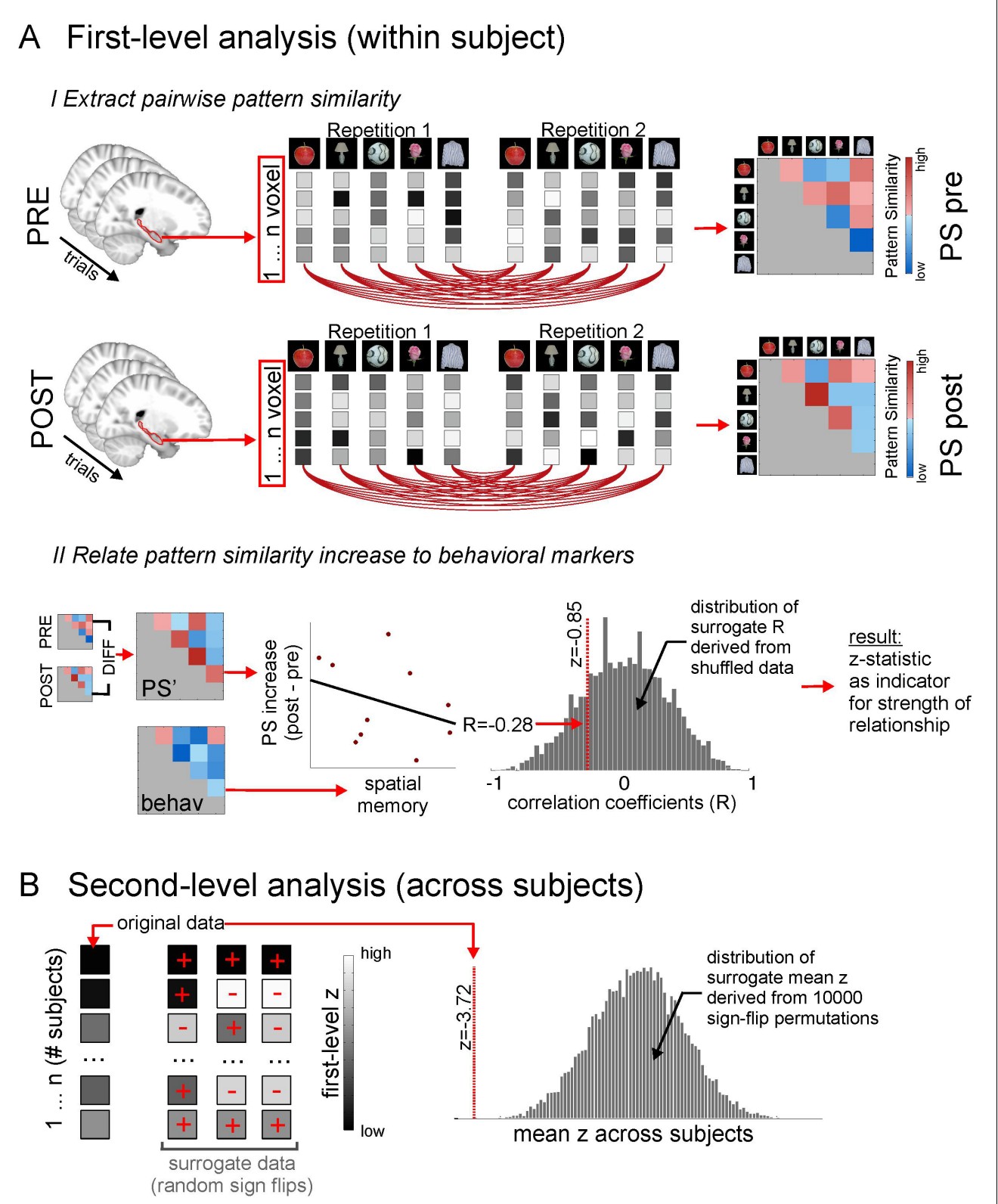

**Figure 4.** Methodological procedure for ROI pattern similarity analysis. (**A**) Illustration of first level analysis. Both for the picture viewing task pre and picture viewing task post, activity of all voxels within a ROI (e.g. bilateral hippocampus) is extracted across all trials, in which 16 different items are presented 12 times (for illustrative purposes, procedures here are depicted for 5 items only). Voxel patterns for every item in every repetition are correlated with voxel patterns for every other item in every other repetition, yielding one average cross-correlation matrix for all items, respectively for

*Figure 4 continued*

the PVT pre and the PVT post task. In the next step, the difference between the PVT post cross-correlation matrix and the PVT pre cross-correlation matrix is formed to get a difference matrix with pattern similarity increases/decreases for every item pair. This difference matrix (PS') is then put in relation to an external variable, for example the remembered spatial distance between every item pair, which is based on the behavioral distance judgment task at the end of the experiment. The relationship between PS' and the external variable is expressed with a correlation coefficient. For example, higher pattern similarity increases for item pairs with lower remembered distance between them (i.e. which were remembered as being closer together) will result in a negative correlation coefficient. To estimate the strength of this relationship, the correlation coefficient is compared to a distribution of surrogate correlation coefficients derived from correlating shuffled pattern similarity increases and distance judgments. The position of the real correlation coefficient in this distribution is a marker for the strength of the effect and is expressed with a z-value, whose absolute value will be higher for more extreme values with regard to the surrogate distribution. However, the z-value can be both positive and negative, depending on which tail of the distribution the real correlation coefficient is located at. (B) Second level analysis. The z-statistics from the first level analysis, which were calculated for every participant, are then tested for significance across participants by comparing the mean z across participants to surrogate mean z-values derived from averaging randomly sign-flipped first-level z-values, with 10,000 repetitions of the random sign-flips. Again, if the mean of the first-level z-values is at an extreme end of the surrogate distribution, this is reflected in a high absolute z-value and a low probability (p) that the effect is not significantly different from zero. See *Figure 4—figure supplement 1* for a corresponding illustration of methodological procedure for the searchlight analysis.

The following figure supplement is available for figure 4:

**Figure supplement 1.** Methodological procedure for searchlight pattern similarity analysis.

However, behavioral analyses had revealed that some participants' spatial and temporal distance judgments were correlated, even though spatial and temporal distances between items were designed to be independent in the task. It could be that the similar effects we find for spatial and temporal memory are only due to the correlation of the two in participants' ratings, i.e. that one of the domains has no unique contribution to the pattern similarity increase. Therefore, we investigated in an additional analysis whether there were separate contributions of the two factors: First, we removed variance explained by spatial distance judgments from the pattern similarity changes in pairs of items in a GLM, and correlated the residuals from this model (i.e. what could not be explained by spatial distance judgments) with temporal distance judgments. We found that these residual pattern similarity changes still correlated with temporal distance judgments in bilateral hippocampus ($Z= -1.805$, $p_{FDR} = 0.041$). Similarly, when we removed the influence of temporal distance judgments first, the residuals still correlated with spatial distance judgments ($Z = -3.719$, $p_{FDR} = 0.0005$). These results suggest that both the dimensions of space and of time contribute to the observed pattern similarity increases.

As outlined above, the main goal of this study was to investigate how the spatial and the temporal aspects of an experience are combined to form a common multi-dimensional event map, into which events can be integrated. Therefore, we correlated the combination of spatial and temporal distance judgments with pattern similarity changes in the hippocampus (i.e., we took the product of the two distance ratings, with the lowest values reflecting proximity in both dimensions and the highest values reflecting high distance in both dimensions). We found that the combination of spatial and temporal distance judgments was indeed associated with hippocampal pattern similarity changes ($Z = -3.719$, $p_{FDR} = 0.0005$). Thus, the spatial and temporal event structures are not only represented separately, but they are also combined in the hippocampus, providing evidence that these two dimensions are flexibly integrated to form a spatio-temporal event map.

Next, we investigated whether right and left hippocampus were differentially involved in representing time and space. Therefore, we computed the five models described above (spatial distance, temporal distance, spatial x temporal combination, temporal distance with effects of spatial distances removed, and spatial distance with effects of temporal distanced removed) separately for voxels in the right and left hippocampus, respectively. We found that pattern similarity across voxels in right hippocampus was significantly correlated with all five factors (all $p_{FDR}<0.041$, significant after FDR correction for 15 multiple comparisons, see Materials and methods). However, for left hippocampus, only spatial distance judgments (with and without effects of temporal distance judgment removed) and the combination of spatial and temporal distance judgments were significantly correlated to pattern similarity changes (all $p_{FDR}<0.011$), while temporal distance judgments were not (neither with nor without effects of spatial distance judgments removed).

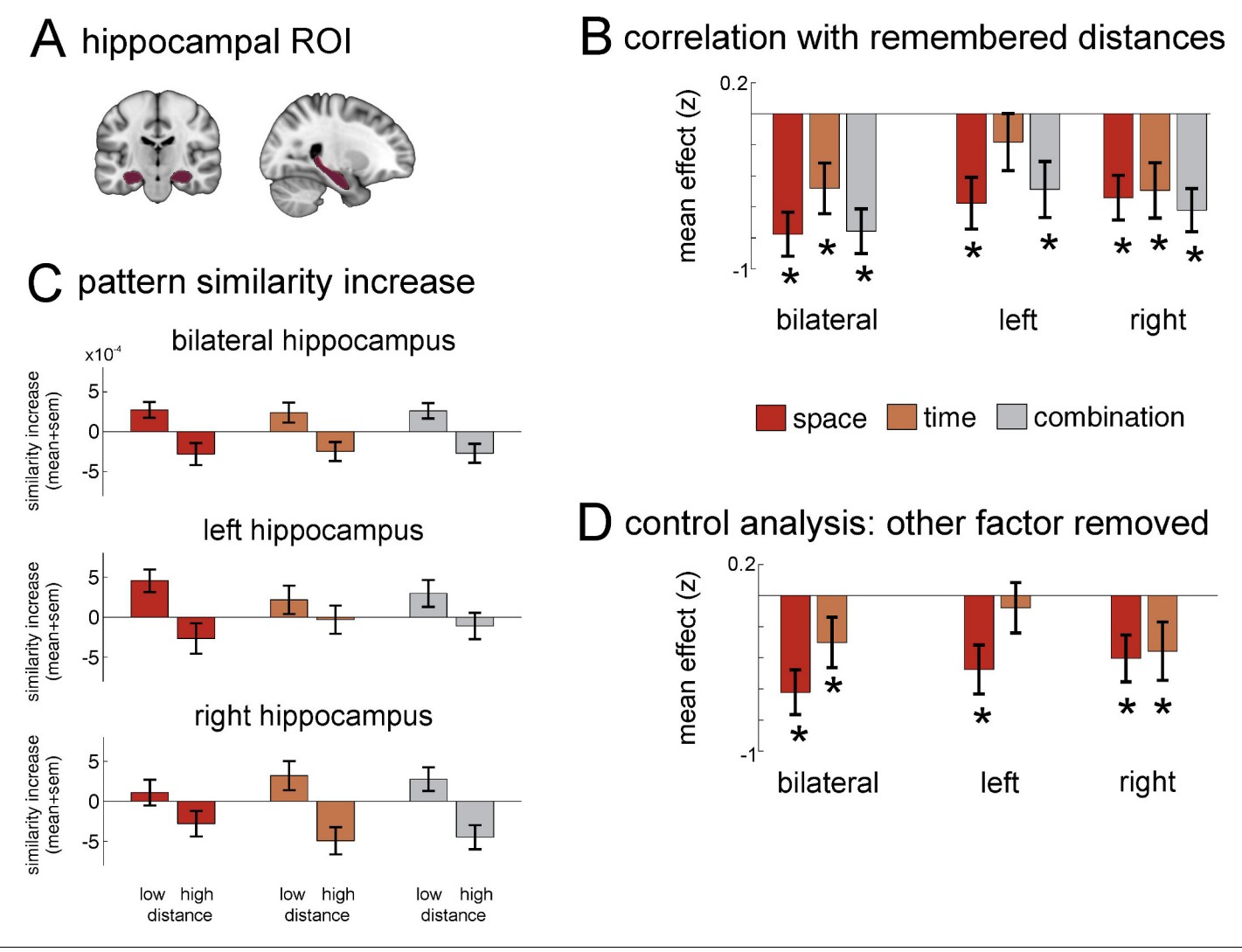

**Figure 5.** Neural similarity of hippocampal multi-voxel pattern scale with spatial and temporal memory and the combination of the two domains. (**A**) Top: Hippocampus mask used for the ROI analysis. (**B**) Increases in pattern similarity (PS') across all grey-matter voxels were negatively correlated with the spatial and temporal distance judgments from the post-scan memory test: The closer together two items were remembered (low distance), the higher was the pattern similarity increase observed in the hippocampus. Results from a bootstrapping procedure are depicted (mean ± sem) for spatial distance judgments and temporal judgments, as well as the combination of both (see Materials and methods for details on analysis). (**C**) Barplots show the averaged pattern similarity increases for item-pairs depending on whether they had low versus high distance to one another in the three conditions remembered spatial distance, remembered temporal distance and the combination of both. (**D**) Because spatial and temporal distance judgments were correlated in the memory judgments, an additional analysis was carried out to calculate the effects after the influence of the additional factor had been statistically removed. Analyses were performed for bilateral hippocampus, as well as for left and right hippocampus separately. Stars in **B** and **D** denote that effects were significantly smaller than zero across participants (statistically corrected for 15 comparisons, see Materials and methods for details on analysis). See **Figure 5—figure supplement 1** for a more detailed ROI analysis of effects on posterior, medial and anterior hippocampus.

The following figure supplement is available for figure 5:

**Figure supplement 1.** ROI analysis of anterior, middle and posterior hippocampus.

## Searchlight analysis of event map-related neural pattern similarity changes

So far, we have provided evidence that the hippocampus is involved in representing and integrating spatial and temporal relationships of multiple events. But are there regions in hippocampus that are

more involved in this effect, and are there any other brain regions that show the same pattern? To address this question, we performed a searchlight analysis over all voxels in our field of view (see Materials and methods). In this approach, a 9 mm sphere is formed around a center voxel and pattern similarity is assessed for all voxels included in that sphere. Moving the center of this sphere consecutively over all possible voxels yields information about fine-grained local effects (see Materials and methods for details).

### Temporal distance

As can be seen in *Figure 6*, a cluster in right medial to anterior hippocampus showed increased pattern similarity effects for objects that were remembered as being temporally close together in the task (peak MNI:26/−18/−22, $T_{25}$ = 4.05, $p_{corr}$ = 0.0178, small volume correction, see Materials and methods). This peak was the global maximum in our acquisition volume and no other effects survived correction for multiple comparisons at a threshold of $p_{corr}$ <0.05. There were no significant effects observed for the opposite contrast (at $p_{corr}$ <0.05). Taken together, these results show that temporal relationships between events in episodic memory are reflected in pattern similarity changes in a cluster in right hippocampus extending from the medial to the anterior part.

### Spatial distance

Next, we performed the same analysis for remembered spatial distances. Again, a cluster in right medial to anterior hippocampus showed the highest pattern similarity increases for objects, which had the smallest spatial distance in the learning task (see *Figure 6*, peak MNI:34/−17/−22, $T_{25}$ = 5.16, $p_{corr}$ = 0.0046, small volume correction). This peak was again the global maximum. No other effect survived correction for multiple comparisons at a threshold of $p_{corr}$ <0.05. There were no significant effects observed for the opposite contrast (at $p_{corr}$ <0.05). The only observed cluster was thus again located in medial to anterior right hippocampus.

## Do spatial and temporal distance judgments have an independent effect?

Our results so far show that partly overlapping regions in right medial to anterior hippocampus represent the spatial and temporal distance between pairs of items. Again, we were interested whether this might be caused by the fact that the two dimensions were correlated in participants' responses during the memory task. As already described in the ROI approach, we removed the influence of the second factor in an additional analysis: Before correlating our PS' matrix with the temporal distance matrix, we removed the effects of spatial distance with a GLM and continued the analysis with the residuals. Conversely, before we correlated our PS' matrix with the spatial distance matrix, we removed the effects of temporal distance with a GLM. We investigated how this procedure affected our searchlight findings in the hippocampus (for peak voxels: MNI 26/−18/−22 for time and 34/−17/−22 for space, respectively). For the temporal structure analysis, the effect was still significant, albeit slightly weaker, after removing the influence of spatial distance ($T_{25}$ = 3.22, $p_{uncorr}$ = 0.0018). The same was true for the peak voxel of the spatial structure analysis after removing the influence of temporal distance ($T_{25}$ = 3.87, $p_{uncorr}$ <0.0004). Thus, part of the overlap in peak regions for space and time may be explained by the two factors being correlated in participants' memory judgments, but there are also significant effects of space and time after statistically removing the influence of the other factor, suggesting that the two dimensions both contribute to pattern similarity increases in right medial to anterior hippocampus.

## Neural changes are modulated strongly by the combination of space and time

Again, we investigated how space and time are integrated to form an event map and assessed correlations between pattern similarity increases and the combined remembered spatial and temporal distances between items by using the product of spatial and temporal distance judgments. We found a significant effect in right medial to anterior hippocampus (see *Figure 6B*, peak MNI: 32/−17/−22, $T_{25}$ = 6.07; $p_{corr}$ < 0.0001, small volume correction). This indicates that pattern similarity in this region increased strongly when items were close together both in the spatial and temporal domain.

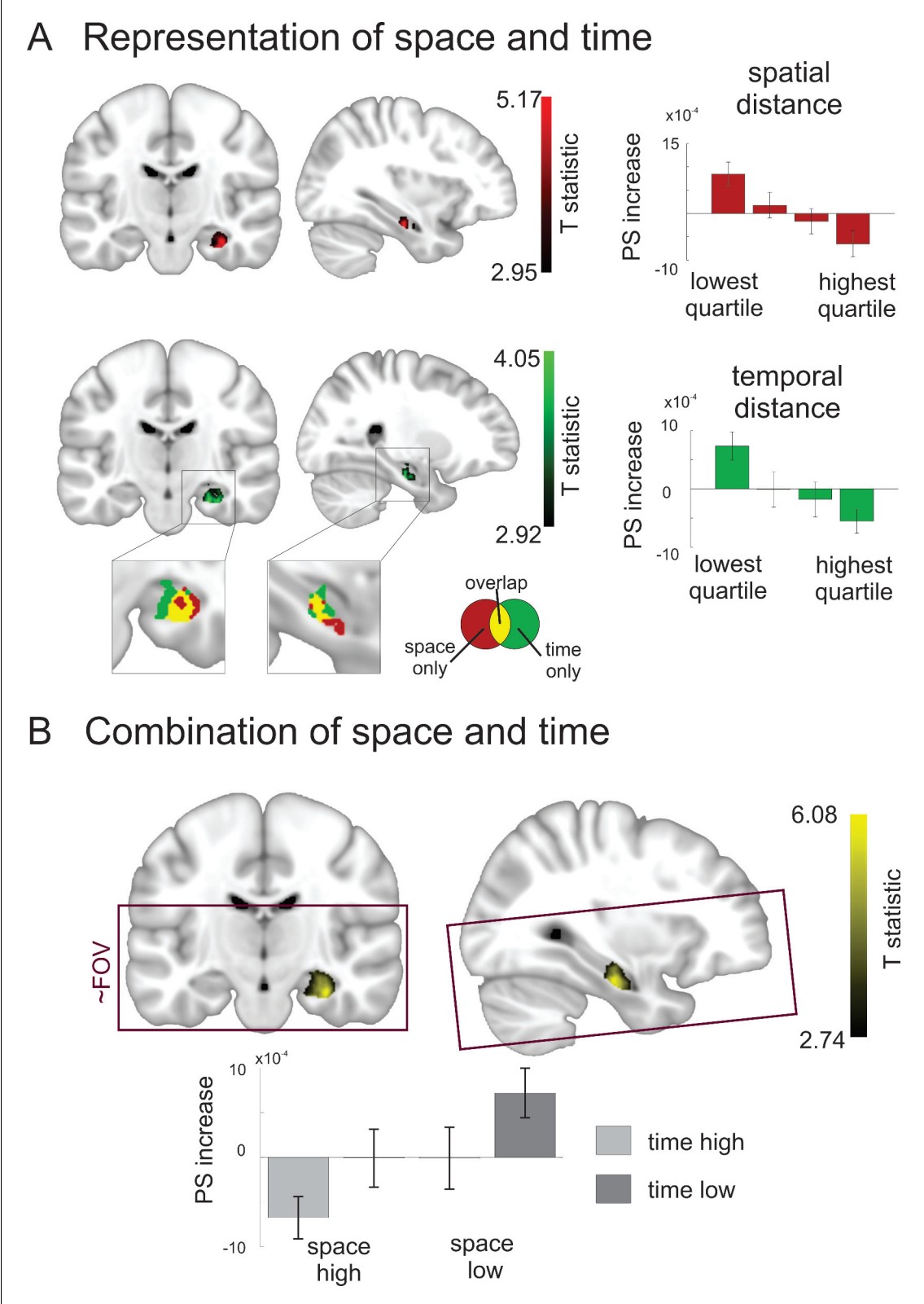

**Figure 6.** Overlapping and distinct codes for spatial and temporal event structures in the hippocampus. Results from the searchlight analysis in which pattern similarity changes in searchlights across the whole MRI acquisition volume were correlated with distance judgments from the post-scanning spatial and temporal memory tests. (**A**) Partly overlapping clusters in right medial to anterior hippocampus show significant correlations between pattern similarity increases and spatial distance judgments, as well as temporal distance judgments; enlarged section of hippocampus shows
*Figure 6 continued on next page*

*Figure 6 continued*

overlapping and separate voxels for the two conditions (binary masks including voxels surviving correction for multiple comparisons of the respective second-level analysis). Bar plots show pattern similarity increases (mean ± sem) for the hippocampal peak separately for different levels of remembered spatial and temporal distance judgments, respectively (memory data binned into quartiles). (B) The effect was strongest when the two factors of space and time were combined and spans the border between medial and anterior hippocampus. Effects are overlaid on a structural template; the color bar indicates T-statistic derived from nonparametric second level analyses (see Materials and methods). Bar plots on the right show parameter estimates (mean pattern similarity increase for the peak voxel). Box indicates approximate field of view (FoV) of the acquisition volume (40 slices at 1.5 mm) for all MR scans. Images are thresholded at $p_{corr} < 0.05$ (small volume corrected, see Materials and methods).

The following source data is available for figure 6:

**Source data 1.** Statistical maps of the searchlight results.

## Impact of objective spatial and temporal distance on pattern similarity

So far, we investigated how the remembered spatial and temporal relationships between items are reflected in pattern similarity increases in the brain. However, in our task there are also objective spatial and temporal relationships between items, independent of how participants remembered them. We defined objective spatial distances between items as the Euclidean distance and the actual temporal distance between items as the median walking time from one item to the next across all repetitions of the route during the navigation task. We then tested the impact of objective spatial distance, objective temporal distance and the combination of the two in an ROI analysis. We found that objective spatial distances between pairs of items were associated with an increase in pattern similarity across all gray-matter voxels in bilateral hippocampus ($Z = -2.85$, $p_{FDR} = 0.02$, corrected for 9 comparisons: 3 ROI $\times$ 3 conditions). Interestingly, no significant effects were found for objective temporal distance and the combination of spatial and temporal distance.

As objective spatial and temporal distances were designed to be independent from one another, we did not need to control for the other factor as we did for the remembered distances. Instead, we made use of the full factorial setup of the objective distances and tested space (high vs low) against time (high vs low) with a 2-way repeated measures ANOVA. We found that only the factor 'space' was significant in bilateral ($F_{1,25} = 8.29$, p=0.008) hippocampus and in left hippocampus ($F_{1,25} = 8.84$, p=0.006), while neither the factor 'time' nor the interaction were significant in any of the three ROIs.

These results suggest that there might be a different pattern of results for objective spatial and temporal distances as compared to remembered spatial and temporal distances. While we think that the remembered distances more accurately reflect the notion of an event map, it would certainly be very interesting to investigate possible differences in the representation of objective distances in future studies, maybe by systematically increasing divergence between objective distances and remembered distances through experimental manipulation.

## Discussion

In this study, we investigated the neural mechanisms underlying the formation of a de novo representation of multiple events embedded in a spatial and temporal context. We used a realistic virtual reality task to induce spatial and temporal interrelations between events and assessed the ensuing change in neural pattern similarity with fMRI. We found that neural similarity in the hippocampus after learning the spatio-temporal event structure scaled with the proximity of event memories in space and time, providing evidence for a mnemonic event map in the hippocampus.

The hippocampal formation is the key region for coding memory for space and time in rodents (*MacDonald et al., 2011*; *Moser et al., 2008*; *O'Keefe and Nadel, 1978*; *Pastalkova et al., 2008*) and humans (*Chadwick et al., 2015*; *Doeller et al., 2010*; *Ezzyat and Davachi, 2014*; *Hsieh et al., 2014*; *Kyle et al., 2015*; *Miller et al., 2013*; *Nielson et al., 2015*; *Schapiro et al., 2016*). It is also essential for forming episodic memory, a hallmark of which is vivid recollection, or 'mental time travel' (*Tulving, 2002*). However, the two mnemonic functions have hitherto mostly been investigated in isolation, even though it has been suggested that the hippocampus supports these different functions via a common underlying mechanism, namely, the representation of multi-dimensional

inter-event relationships in memory space (*Eichenbaum, 2014*; *Eichenbaum et al., 1999*). Here, we take a first step towards demonstrating such a common coding mechanism in the human hippocampus by showing that both spatial and temporal relationships between events might be represented by a similar mechanism.

Furthermore, our study goes beyond previous findings in several important ways: Firstly, we investigate the de novo formation of a spatio-temporal event map by tracking the task-induced changes in pattern similarity from a pre-task baseline scan to the post-task scan. While this approach enables us to control for the potential effects of stimulus and task confounds (see below), most crucially it allows us to investigate representational changes (through comparison of post- vs pre-acquisition effects) as a consequence of encountering a complex event structure during the learning phase.

Secondly, we directly relate the specific neural changes we observe to the interrelations of the memories that have been formed. We achieve this by mapping out the participant-specific mnemonic event-map for the newly acquired spatio-temporal structure in an extensive post-scanning memory test. Previous studies have related the strength of neural effects to markers of overall memory performance across participants (*Hsieh et al., 2014*), or have restricted their analyses to trials with self-reported recollection success (*Nielson et al., 2015*). One study reports higher pattern similarity during task trials for items which were later judged as 'close' as compared to items which were later judged as 'far' (*Ezzyat and Davachi, 2014*). However, all of the item pairs which entered the analyses were, in fact, separated by the same number of intervening trials in the task (i.e., two trials), limiting the complexity of the probed memory. Here, we probed all possible interrelations between the encountered events across spatial and temporal dimensions simultaneously. This allowed us to reconstruct, from these pairwise ratings, the full participant-specific temporal and spatial distance maps which were then used in the representational change analysis of our fMRI data.

Thirdly, we combine spatial and temporal aspects in our learning task and use teleporters to reduce overlap between spatial and temporal distances. This makes the task more complex and increases the level of abstraction required to accurately represent the events. Notably, all item pairs are defined by a specific spatial as well as a temporal distance. Thus, for solving the memory task, it is not sufficient to have a notion of two items belonging together or being close, but one needs to retrieve their spatial *and* temporal position in the task and estimate how close they are in the respective domain separately. It should be noted that space and time were intertwined to some degree in participants' memory reports. However, the generally good fit between responses and actual distances and results from the additional analyses in which we statistically control for the influence of the other factor indicate that participants were able to represent the two dimensions separately, at least to a certain degree.

## A novel approach to investigate the neural structure of a spatio-temporal event map

In this study, we introduce a novel experimental paradigm that allows us to investigate both spatial and temporal aspects of memory and combines a complex episode-like learning task with rigorous experimental control. So far, most studies in the field have investigated the neural underpinnings of memory either for space (*Bellmund et al., 2016*; *Doeller et al., 2010*; *Iglói et al., 2010*; *Kyle et al., 2015*; *Vass and Epstein, 2013*; *Wolbers et al., 2007*) or for time (*Ezzyat and Davachi, 2014*; *Hsieh et al., 2014*). However, in real life, episodes are always embedded in both spatial *and* temporal context, as recently demonstrated by a study using GPS and camera timestamps of snapshots of real-life experiences of participants to show that pattern similarity in the hippocampus is sensitive to spatial and temporal distances over large scales of magnitude (*Nielson et al., 2015*). We include both spatial and temporal aspects in our learning task to test, in a laboratory setting, how the two dimensions are represented after the de novo formation of an event map.

Another aspect of our task is that it induces spatio-temporal memories by exposing participants to a realistic, 3D virtual environment. This paradigm is well suited to mimic episodic memory formation, due to both the richness of experience and the active nature of the task, in which participants have control and agency over the to-be-encoded events. Volitional control is a crucial aspect of the hippocampal role in memory encoding (*Voss et al., 2011*) and a sense of self, in turn, may be an essential prerequisite for mental time travel (*Tulving, 2002*).

However, in a complex, realistic learning task, analyses might be prone to potential confounds. For example, items that are spatially proximal in the task (or in real life) probably also share a similar view, especially in the seconds walking up to the item, and would therefore inflate pattern similarity. Likewise, items that are temporally proximal would automatically have higher neural pattern similarity due to autocorrelations in the slow BOLD signal. To counter these potential confounds, we limit our fMRI analysis to the difference in pattern similarity between two separate blocks, which were scanned both before and after the task – an approach taken in several recent fMRI studies (*Collin et al., 2015*; *Milivojevic et al., 2015*; *Schapiro et al., 2012*; *Schlichting et al., 2015*). One strength of this approach is that by strictly focusing on the change in pattern similarity from PVT pre to PVT post we can exclude effects of temporal proximity between items in the PVT tasks and of a priori differences in neural pattern similarity that some pairs of items might elicit in individual participants, an aspect that cannot easily be excluded by using pairs of autobiographical photographs, for example.

Taken together, the combination of these advances gives us access to study the spatio-temporal organization of memory in humans – spanning a triad between an experimentally created objective 'external world' as simulated with our realistic, life-like task, a subjective representation of this external world in participants' minds as assessed with the extensive memory testing, and the investigation of how this subjective representation is reflected in the brain as expressed in changes in neural patterns associated with an event map.

## Mechanisms underlying spatial and temporal memory

Are similar or different neural mechanisms supporting spatial and temporal memory? For space, the existence of place cells (*O'Keefe and Dostrovsky, 1971*) and grid cells (*Hafting et al., 2005*) has suggested that space is represented in an abstract manner, potentially in the form of a 'cognitive map' (*O'Keefe and Nadel, 1978*). In contrast, the representation of temporal structure has been discussed more in terms of analogous mechanisms: chaining models (*Axmacher et al., 2010*; *Jensen and Lisman, 2005*) argue that serial events are linked through pairwise binding between succeeding items (through LTP-like mechanisms) and that the recall of one item triggers recall of the subsequent item. The temporal context model (*Howard and Kahana, 2002*; *Howard et al., 2005*) suggests that an episodic element is 'tagged' to slowly changing, random neuronal background activity present at the time of encoding; this temporal context is then reinstated during recall and provides information about how long ago the episode was experienced by assessing the degree of disparity between the reinstated and the present neuronal background (i.e. the greater the disparity, the more time has passed). In both of these models, temporal structure in memory can be seen as a mere by-product of basic neuronal processing. However, the recent findings about internally generated sequential firing of neuronal ensembles (*Pastalkova et al., 2008*) and context-specific time cells (*MacDonald et al., 2011, 2013*) are consistent with the notion of a more active mechanisms in temporal memory, which might, in fact, be very similar to mechanisms in spatial memory (*Howard and Eichenbaum, 2015*). In humans, it has also been shown that hippocampal damage leads to impairments in both spatial and temporal memory tasks (*Spiers et al., 2001*; *Konkel et al., 2008*) and that the hippocampus is active during active retrieval of temporal sequences as well as spatial layouts (*Ekstrom et al., 2011*), even though dissociable networks for the two retrieval domains were observed outside of the hippocampus. In seeming contrast with our results, one study investigating pattern similarity during retrieval of spatially near versus far intervals and temporally near versus far intervals found an interaction effect in right hippocampus, with increased pattern similarity for spatially far compared to spatially near retrieval trials and the opposite effect for temporally near versus far retrieval trials (*Kyle et al., 2015*) whereas we find a pattern similarity increase both for spatially close and temporally close pairs of items. However, this discrepancy can probably be explained by methodological differences: we recorded our data not during active retrieval, but during two independent tasks in which participants had to passively view items, and then related the pattern similarity difference between these two tasks to an external behavioral marker, i.e. participants subjective distance ratings, which was collected at a later time point. Another study found decreased pattern similarity in hippocampal subfields CA2/CA3/DG between trials in a spatial retrieval condition and a temporal retrieval condition when both domains were correctly retrieved compared to trials when only one of the domains was correctly retrieved, and the opposite effect in parahippocampal cortex (*Copara et al., 2014*). Again, it is difficult to directly relate this study to our results, since the pattern

similarity differences were found in a retrieval task and in hippocampal subfields, which we did not investigate in this study. However, it would be interesting to also acquire data during the memory test to examine whether actively retrieving spatial and temporal relationships affects neural pattern similarity. Our data support the notion of a common hippocampal coding mechanism in space and time: neural similarity scales with the proximity of event memories in both dimensions. Notably, while we found that spatial and temporal distance are to some degree correlated in participants' memory, the observed effect is still present for each domain after statistically controlling for the effect of the other domain, suggesting that both space and time contribute to the observed pattern similarity increase, possibly in an additive manner. The observed strong effect for the combination of space and time further suggests that the two dimensions might be integrated in a common dimension in the hippocampus, i.e. spatio-temporal proximity, supporting the formation of hierarchical structures in a memory space (*Collin et al., 2015*; *Eichenbaum et al., 1999*; *McKenzie et al., 2014*). Here we provide evidence for a mapping of the entire event structure in the hippocampus. The hippocampal event-coding patterns are thus not restricted to representing event relationships per se, but rather scale with mnemonic distance in a spatio-temporal event map. One alternative explanation for the increased pattern similarity could be that in the second PVT participants covertly retrieve the environment in which they encountered the object in the learning task and that the effect might be partly related to the higher visual similarity in the imagined scene. Several points argue against this interpretation. Firstly, the views associated with nearby boxes are not necessarily very similar, for example due to rotations during navigation between the boxes or large buildings obstructing the view at one location but not the other. Secondly, we observe the increased similarity for temporally close items as well, which due to the teleporters are not necessarily spatially close. Thirdly, if the effects relied solely on similarity in visual scenes, we would expect to see very prominent pattern similarity increases in visual areas. However, no cluster survived correction for multiple comparisons outside of hippocampus. Therefore, we believe that our findings reflect memory for spatial and temporal relationships, rather than visual similarity.

One interesting finding here is that we observed a different pattern of results regarding the neural representation of objective spatial distances compared to remembered distances. It is conceivable that the spatial and temporal distances as they are remembered are more indicative of the event map which participants have formed, but the different pattern of results for the objective distances raises the interesting question how objective distances are translated into subjectively remembered distances, and how this is reflected in the neural representation. In our behavioral analyses we found that memory judgments in one domain were biased by the distances in the other domain, but no domain seemed to have a higher impact than the other. It is very likely that other factors in addition to objective spatial and temporal distance impact how a spatio-temporal event map is constructed and remembered, and it will be very interesting in future studies to identify these factors.

## Conclusion and outlook

By showing that both the temporal and the spatial relationships between multiple events are represented in the hippocampus, we took a first step towards unraveling the link between the multi-faceted external world, participants' memories of it and the neural coding mechanisms supporting the formation of a multi-dimensional mnemonic structure. Such event maps are likely not restricted to the physical dimensions of space and time. Elements in memory could be arranged according to a variety of factors, for example social aspects (*Kumaran and Maguire, 2005*; *Kumaran et al., 2012*; *Tavares et al., 2015*) or abstract concepts (*Milivojevic and Doeller, 2013*). It would be interesting to investigate in further studies whether nuanced differences in these dimensions can be read out in hippocampal patterns as well. Another exciting future avenue for research could be that – if mnemonic relatedness in participants' minds is reflected in pattern similarity – one can reverse the logic and use participant-specific similarity maps to make inferences about their internal record of experience (*Morris and Frey, 1997*; *Wood et al., 1999*). In summary, the present study sheds light on the neural mechanisms supporting the formation of a spatio-temporal event map in memory by leveraging a novel, life-like learning task in combination with rigorous experimental control. More broadly, it may be a first step towards mapping out the representation of the external world in the human mind – here along physical dimensions, but potentially also along more abstract ones.

## Materials and methods

### Participants

Based on an effect size of d = 1.03, which was found in a previous similar study from our lab (*Milivojevic et al., 2015*) in hippocampus, an alpha level of 0.001 (necessary for corrections for multiple comparisons in fMRI data) and power of 0.95, a sample size of N = 26 was calculated to be necessary using G*Power (http://www.gpower.hhu.de/). 26 participants signed up for the study through a University-wide online recruitment system. The mean age of the group was 24.88 ± 2.21 (mean ± std) and 11 were female. All participants underwent a familiarization phase in Donderstown, so they had good knowledge of the city (see below). All participants gave written informed consent, filled in a screening form to ensure they did not meet any exclusion criteria for fMRI and were compensated for their time. The study was approved by the local ethics committee (CMO Regio Arnhem-Nijmegen).

### Virtual city environment 'Donderstown' and familiarization phase

For the purpose of providing a realistic, life-like episodic learning experience, we developed a 3D virtual city environment using the Unreal Development Kit for Unreal Engine 3 (https://www.unreal-engine.com/previous-versions). The city consists of a complex network of streets and features residential as well as commercial areas. Distances in the VR city are difficult to translate to real-world settings, because they are based on arbitrary units that depend on the exact scaling of the 3D meshes. Relating the eye-level height of the first-person player (assumed to be at 1.60 m) to these arbitrary units, one side length of the square city roughly translates to 390 m. Walking this side length takes approximately 36 s, putting the walking speed with 39 km/h well above normal walking speed. However, this was necessary to achieve a sufficient number of repetitions in reasonable time. See *Figure 1* for a top-down overview of Donderstown and http://www.doellerlab.com/donderstown/ for further images. In the experiment, participants had to navigate through this complex virtual environment and judge distances in it (see below for details on the task). During piloting of this study we observed that it was difficult for most participants to accurately estimate Euclidean distances if the city environment was novel to them. Some participants showed signs of disorientation, especially with regard to those parts of the route in which they were teleported through the city (see below). As our hypotheses depended crucially on the use of these teleporters, we decided to only include participants with extensive prior knowledge of the city. Thus, all participants were required to have taken part in another study from our lab, which pre-exposed them to the virtual city, Donderstown. In this previous study, which took place on a different day (1–21 days prior to participation in the current study), participants had to learn the names and locations of specific houses in this city and estimate directions between these houses. The task was unrelated to the current task but exposed participants to Donderstown for approximately 2 hr and thereby ensured that they had formed a robust spatial representation of the city. This experimental session was crucial to ensure successful learning of the spatio-temporal trajectories. Notably, participants did not acquire any knowledge about the position of the wooden boxes or teleporters (see *Figure 1* and below), as neither was present in the familiarization task.

### Experimental sessions

The experiment consisted of five parts (see *Figure 3*), two of which took place in the MRI scanner. First, participants were asked to freely navigate through Donderstown for 10 min to refresh their knowledge of the city. This session was performed in front of a computer screen outside of the scanner. Secondly, participants were taken inside the scanner and performed the picture viewing task ('PVT pre'), in which 17 objects were presented 12 times in random order. Thirdly, they were taken out of the scanner and performed the route learning task in front of a computer screen in a behavioral lab. In this learning task, 16 objects were arranged at specific locations along a route and the spatial and temporal distances between them were varied independently in a 2-by-2 design. After the learning task, participants were taken inside the scanner again for the picture viewing task ('PVT post'), in which 17 objects were presented 12 times in the same random order as before. Lastly, participants performed three different memory tests outside of the scanner: a free recall test, a distance judgment test and a map test. All of the tasks are described in detail below.

## Spatio-temporal learning task

The task is similar to playing a computer game and involves first-person navigation through a 3D virtual city environment, Donderstown. In the learning task, participants started at a specific point in the city and then had to follow a predefined route through the city. At the beginning, participants were unfamiliar with the route, therefore it was marked by presenting orange traffic cones in regular intervals. Participants' task was to follow the traffic cones until they arrived at a wooden box; then they were required to touch the box. The box then opened and the content of the box was revealed by presenting a single object on a black background. After 2 s, the black screen with the object disappeared and participants continued to follow the route marked by the traffic cones. Participants encountered 16 different objects along the route, always hidden in a wooden box. Pictures depicted various every-day objects (e.g. a football, apple), the requirement being that they would reasonably fit inside the wooden box. The next box (and the traffic cones leading up to it) would only appear after the previous box had been opened (i.e. touched).

Crucially, at specific points during the route, participants encountered a teleporter after opening a box. When they touched this teleporter, they were transported immediately to a completely different part of the city, where they would encounter the next box to be opened. These teleporters were always in the same position along the route and created experimental situations in which the next box was opened after only a small temporal delay while maintaining long spatial distance between the two boxes. This was necessary for rendering the two factors of time and space independent of each other (see *Figure 1—figure supplement 3* for a comparison of spatial and temporal distances; also note that in some participants' memory, the two factors were not independent from one another, but correlated). After participants opened the last box in the route, a black screen appeared for 15 s and participants found themselves back at the start of the route, where they again followed the orange traffic cones until they found the first box in the route again.

A wooden box at a particular position during the route always contained the same object for a given participant (between participants, the content of the wooden boxes was randomized). Thus, an object was associated with a particular position in Donderstown, and it was always encountered in the same temporal order. The route was taken by participants 14 times in total.

After 6 repetitions, the orange traffic cones were no longer shown and participants had to find the route on their own. During piloting, it had become evident that participants tended to underestimate their dependence on the traffic cones and would sometimes be surprised by the difficulty of navigating in the absence of the cones. Therefore, we included an 'emergency help' procedure that was active for 5 repetitions after we removed the traffic cones. When participants felt they had got lost, they were instructed to return to the position of the last box or the last place they were certain was on the correct route and press the button 'H' on the keyboard. On this, all of the traffic cones leading up to the next box appeared at the same time and gave participants an opportunity to find their way again. As this was a free navigation task, duration for the completion of the task varied considerably between participants, lasting 71.63 ± 13.75 (mean ± std) minutes. In summary, the learning task in this study was developed to induce a spatio-temporal structure between different objects by presenting them repeatedly and consistently at a specific place and in a specific temporal order.

## Memory tasks

Memory was assessed with three different tasks, (1) a free recall task, (2) a spatial and temporal distance judgment task and finally a (3) map task:

During the *free recall task*, participants named all objects, which they encountered during the learning task using a microphone. They were given two minutes in this task and were instructed to name the objects in the order in which they came to their mind.

In two *distance judgment tasks*, participants were asked to rate the spatial and temporal distance between pairs of objects (see *Figure 1B*). This task comprised 240 trials and lasted 45–70 min (depending on participants' speed). There were two conditions: one, in which participants were required to rate the distance between objects with regard to their Euclidean spatial distance (*spatial memory task*), and secondly, to rate the distance between objects with regard to the time it had taken to walk from one object to the other (*temporal memory task*). During every trial, participants were shown two objects that they had seen in the learning task and then made their distance

judgment by sliding a bar with the mouse on a range from 'close together' to 'far apart'. Participants were instructed to base their rating on the smallest and largest distance that was present in the task, i.e. the smallest distance in the task would correspond to the bar position closest to the 'close together' end of the scale whereas the largest distance in the task would correspond to the 'far apart' end of the scale. To make the task easier for participants, conditions were sampled in 8 blocks: 30 trials of one condition were shown in one block, then there was a break of 20 s and then the next block would start with 30 trials from the other condition. Before each block, the condition of the next block was shown and participants had to press a button to continue. In addition, either the cue 'space' or 'time' was displayed in every trial above the pair of objects to be rated. Whether the test started with the 'time' or the 'space' condition was counterbalanced across subjects. Participants were explicitly instructed that the spatial and temporal distance ratings were in some cases quite different from one another. The instruction explicitly mentioned that items could still be close together in space even when they were at different ends of the route (i.e. far apart in time) due to the route leading back and forth through the city, and it was also pointed out that items could be far apart in space but close together in time due to the teleporters. Notably, this instruction was only given to participants immediately before the distance judgment task, i.e. after the imaging part was already concluded, so the effects we find in the fMRI data cannot be explained by this explicit instruction.

The final *map test* lasted approximately 5 min and involved 16 trials. In every trial, participants were shown one of the 16 objects encountered in the city. Then, they saw a schematic aerial view of Donderstown and had to indicate where in the city the object had been located by moving the mouse to the memorized location. To make sure participants could translate their first-person perspective of the city during the learning task to a topdown view, certain prominent landmarks in the city were marked with symbols and pointed out to them before the start of the memory task.

## Picture viewing task during scanning

Before (picture viewing task, PVT pre) and after (PVT post) the learning task, pictures of 17 different objects (see *Figure 1—figure supplement 1*) were presented repeatedly on a black background. 16 of these objects were used in the learning task, while 1 object only served as a target and was not shown during learning. Whenever participants saw this target, they were asked to press a button to ensure that they attended the stimuli throughout the PVT pre and post blocks. In 25 participants, the target was detected in 98.33 + 4.17 percent of cases in both PVT pre and PVT post blocks. Due to malfunctioning of the button box in the PVT post block target detection data from one participant could not be recorded.

During each picture viewing task, every object was shown 12 times, once in each of 12 blocks in pseudo-random order (see below). Between blocks, there was a 30 s break. In total, each of the two picture viewing tasks took 23 min and included 204 trials. In every trial, participants saw the picture for 2500 ms, followed by a fixation cross until the next trial started (intertrial interval, ITI). The next trial commenced either two or three TRs after the start of the previous trial (2 TRs in 50% of trials; 3 TRs in the other 50% of trials, with both types of inter-trial intervals, ITIs, randomly assigned to trials).

Since we were interested in the changes in pattern similarity that were the result of the learning task (and not due to spurious timing differences between the PVT pre and PVT post), we used identical PVT pre and PVT post blocks for every participant. More specifically, we created one 'recipe' for every participant. This recipe described which object was shown in which trial and also defined the distribution of the different ITI types. While the recipe for each participant was created to be semi-random (see below) and was, in fact, different for every participant, we re-used the recipe from PVT pre for PVT post within a participant, thereby ensuring that the two tasks were absolutely identical. In theory, participants could have realized that the order in the second task was repeated from the first, but as there were 204 trials, it seems unlikely that they remembered sequences of multiple items. In fact, none of the participants reported to have noticed a specific order.

For creating the recipe, we took into account that every object was supposed to be shown only once in every block. Therefore, we shuffled the order of the 17 objects 12 times and concatenated the resulting vector. Then, we assigned an ITI of either 2 or 3 TRs to every trial, balancing the occurrence across the entire experiment. Lastly, we performed a one-factor ANOVA (with the 17 different

objects as groups) on the positions of every object in this recipe. If this ANOVA was significant (meaning that the positions of at least one of the objects were consistently early or late in the blocks), we discarded the recipe and created a new one. This measure was taken to prevent huge imbalances in the ordering of the objects. As we always compared pattern similarity from the pre and the post blocks in our analysis (see below), any effects that are solely due to spurious order effects should be present in both sessions and therefore cannot explain differences in neural pattern similarity.

## Image acquisition

MRI data were collected on a 3T Siemens Skyra scanner (Siemens, Erlangen, Germany). A high-resolution 2D EPI sequence was used for functional scanning (TR = 2270 ms, TE = 24 ms, 40 slices, distance factor 13%, flip angle 85 degree, field of view (FOV) 210 × 210 × 68 mm 3, voxel size 1.5 mm isotropic). The field of view (FOV) was aligned to fully cover the medial temporal lobe, parts of ventral frontal cortex and (if possible) calcarine sulcus. Functional images for PVT pre and PVT post were acquired in separate sessions. In addition to these partial-volume acquisitions, 10 scans of a functional wholebrain sequence were also acquired (usually in both sessions, but due to time pressure sometimes only in one session) to improve registration during preprocessing. The sequence settings were identical to the functional sequence above, but instead of 40 slices, 120 slices were acquired, leading to a longer TR (6804.1 ms). A 0.8 mm structural scan was acquired for every participant (TR = 2300 ms; TE = 315 ms; flip angle = 8°; in-plane resolution = 256 × 256 mm; number of slices = 224, voxel resolution = 0.8 × 0.8 × 0.8 mm3). Lastly, a gradient field map was acquired (for N = 21 participants), with a gradient echo sequence (TR = 1020 ms; TE1 = 10 ms; TE2 = 12.46 ms; flip angle = 90°; volume resolution = 3.5 × 3.5 × 2 mm; FOV = 224 ×224 mm).

## fMRI preprocessing

Preprocessing of functional images was performed with FSL (http://fsl.fmrib.ox.ac.uk/fsl/fslwiki/). Both the main functional scans and the short wholebrain functional scan were submitted to motion correction and high-pass filtering at 100 s. For those participants with a fieldmap scan, distortion correction was applied to both functional data sets. No spatial smoothing was performed. The two functional datasets (PVT pre and PVT post) were then both registered to the preprocessed mean image of one wholebrain scan (if a wholebrain scan was acquired in both sessions, the first wholebrain scan was used for both). This was done to ensure that voxels from these two separate sessions were corresponding to the same anatomical location. The two brain masks from the PVT pre and PVT post blocks were also registered to the wholebrain space and intersected: only voxels which were covered in both sessions were analyzed during the next step. The whole brain functional images were registered to the individual structural scans. The structural scans were then in turn normalized to the MNI template (at 1 mm resolution). Grey-matter segmentation was done on the structural images and the results were mapped back to the space of the wholebrain functional scan for later use in the analysis.

## Representational similarity analysis

Representational similarity analysis (RSA; *Kriegeskorte et al., 2008*) was carried out separately for the PVT pre and PVT post blocks. The preprocessed scans were loaded into Matlab as 4D matrices. For every voxel, movement correction parameters were used as predictors in a GLM with the voxel time series as dependent variable. The residuals from this GLM (i.e. what could not be explained by motion) were then taken to the next analysis step. As the presentation of images in the PVT pre and post blocks was locked to the onset of a new volume (see above), the third volume after image onset was selected for every trial (effectively covering the time between 4540–6810 ms after stimulus onset). Only data for the 16 objects that were shown in the city were analyzed, discarding data for the target object. Data were then sorted according to object identity and repetition, yielding a 16 × 12 matrix for every voxel (16 objects, 12 repetitions). Resulting data were then subjected to two different types of analyses, (1) a region of interest (ROI) based analysis and (2) a searchlight analysis.

## ROI analysis

Hippocampal masks were created using the hippocampal ROIs of the probabilistic Harvard-Oxford atlas provided by FSL (*Desikan et al., 2006*; *Makris et al., 2006*), thresholded at a probability level of 0.25. We generated one mask for bilateral hippocampus, one for only the left, and one for only the right hippocampus. These masks were then coregistered to the subject specific functional space. Then, the masks were intersected with the subject-specific grey-matter masks, leaving only grey-matter voxels in the hippocampus. The trial-wise values for every voxel within this mask were then extracted and the voxel pattern for every object in every repetition was correlated with the voxel pattern of itself *and* every other object in every other repetition. Thus, every trial was correlated with every other trial, except combinations of trials within the same of the 12 blocks. Mean correlation coefficients for every possible pair of objects across repetitions were calculated, yielding a 16-by-16 cross-correlation matrix for every ROI and every PVT block. Subsequent analysis of this cross-correlation matrix (see below) was identical for the ROI and searchlight approach.

## Searchlight analysis

Instead of including all voxels within an anatomically defined region, all voxels in a sphere around a given voxels were studied in the searchlight analysis, allowing a more regionally specific analysis in the entire field of view. Around every voxel of the subject-specific combined brain mask, a sphere was formed with a radius of 6 voxels (9 mm). Within this sphere, only grey-matter voxels were considered. If less than 30 voxels remained, the searchlight was not analysed further. Within every valid searchlight, the approach was analogous to the ROI analysis: the voxel pattern for every object in every repetition was correlated with the voxel pattern of itself and every other object in every other repetition. So, again, every trial was correlated with every other trial, except combinations of trials within the same block. Mean correlation coefficients for every possible pair of objects across repetitions were calculated, yielding a 16-by-16 cross-correlation matrix for every searchlight.

## Analyzing the cross-correlation matrices

The 16-by-16 cross-correlation matrices for every ROI and every searchlight reflected the mean pattern similarity between pairs of objects. We calculated similarity matrices separately for the PVT pre and the PVT post block, reflecting the neural response to the objects when participants had not seen them in the virtual city context, the other reflecting the neural response to the objects after the spatio-temporal relationship between objects had been learned. Subtracting the PVT pre similarity matrix from the PVT post matrix resulted in a matrix that reflected the change in pattern similarity for all pairwise comparisons of items that is due to the learning task. In a second step, the pattern similarity difference matrix (PS' matrix) for every ROI and every searchlight was related to an external variable which was derived from the behavioral tasks, for example the participant-specific remembered spatial distances for all pairs of items, which were derived from the distance judgment that was performed outside of the scanner at the end of the experiment. In our main analyses, we correlated the PS' matrix with the remembered spatial and temporal distances as derived from participants' responses in the memory test, as well as with the combination of the spatial and temporal distance, using Spearman correlation. The size of the resulting correlation coefficient describes the fit between the PS' matrix, i.e. the pattern similarity increase from PVT pre to PVT post, and a given variable. For the searchlight analysis, the correlation coefficient describes the relationship only for the given searchlight, and it is assigned to the center voxel of the searchlight. Iterating through all searchlights in the field of view, this results in a brain map of correlation coefficients for a participant that can then be taken to second-level testing (see *Figure 4—figure supplement 1* for an illustration of this). For the ROI analyses, the correlation coefficient reflects the relationship between pattern similarity increase and the external variable for the entire ROI. Here, we applied an additional bootstrapping procedure to estimate the strength of the relationship (for the searchlight approach, this would have been computationally too demanding). For this, we shuffled the two matrices, which were used to compute the correlation coefficient (PS' and the external variable) against one another, so that the relationship between them was random. We then calculated a Spearman correlation for these shuffled data and repeated the procedure 10,000 times. Thus, for every participant, we gained a surrogate distribution of Spearman correlation coefficients, which were based on shuffled data and compared our real correlation coefficient for a given ROI against this random distribution (see

*Figure 4* for an illustration of this procedure). To this end, we gained a z-statistic for every participant, reflecting how normal or extreme the real correlation coefficient was when compared to the random distribution. The resulting z-statistic was then tested across participants for the ROI analyses.

## Pattern similarity control analyses

Because temporal and spatial distance estimates were correlated in some participants' behavioral ratings, it could be that effects we find for one domain can be explained by the correlation with the other domain, and that the domain has no unique contribution to the effect we find. Therefore, we performed a control analysis (both for the ROI analysis and for the searchlight analysis) in which we investigated whether there were unique contributions of the two domains: Before correlating pattern similarity increases with the remembered temporal distances, we performed a GLM in which we entered remembered spatial distance as a predictor and pattern similarity increases as criterion. Then, we took the residuals of this GLM (i.e. variance not explained by remembered spatial distance) and correlated these residuals with the remembered temporal distances. Conversely, before we correlated pattern similarity increases with the remembered spatial distances, we performed a GLM modeling the effects of remembered temporal distances and calculated the correlation with remembered spatial distances with the residuals of this analysis. We then took these correlation coefficients to the same second-level analysis as described below for the other analyses.

## Second-level testing of RSA results

For the ROI analysis, the z-statistics were averaged across participants and tested for significant deviance from zero using a non-parametric approach: the observed mean z-statistic was compared to a distribution derived by performing random sign-flips on the participant-specific values and averaging them. This was repeated 10,000 times, yielding a null distribution of average z-statistics with random signs. Then, the real average was tested against this distribution of random averages and the resulting z-statistic reflects how much the real average deviates from the random distribution. This procedure closely follows methods applied in other studies using Representational Similarity Analysis (*Schlichting et al., 2015*; *Stelzer et al., 2013*). For the ROI approach, we also corrected for 15 multiple comparisons (3 ROI × 5 effects) by applying false discovery rate correction (*Benjamini and Yekutieli, 2001*) to the probability values from the non-parametric second-level test; reported p-values are FDR corrected.

For the searchlight analysis, the resulting correlation coefficient brain maps were tested in a second level analysis to identify regions in which correlation coefficients consistently differed from zero across participants using a non-parametric equivalent of the one-sample t-test implemented in the randomise package provided by FSL (http://fsl.fmrib.ox.ac.uk/fsl/fslwiki/Randomise; *Winkler et al., 2014*), using 5000 random sign-flips and threshold free clustering. Results as reported above and denoted with pcorr are corrected with FSL-randomise with a small volume correction for the bilateral hippocampus, based on the same hippocampus mask that was used in the ROI approach. Note that in both the ROI and the searchlight approach, we tested whether correlation coefficients were consistently negative as we expected *low* spatial or temporal distance to be associated with *high* pattern similarity in the brain.

## Acknowledgements

This work was supported by the European Research Council (ERC-StG 261177) and the Netherlands Organisation for Scientific Research (NWO-Vidi 452-12-009 and NWO-MaGW 406-14-114). The authors would like to thank A Vicente-Grabovetsky and B Steemers for help with programming the virtual navigation task and B Milivojevic and S Collin for helpful suggestions on the manuscript.

## Additional information

### Funding

| Funder | Grant reference number | Author |
|---|---|---|
| European Research Council | ERC-StG 261177 | Christian F Doeller |
| Nederlandse Organisatie voor Wetenschappelijk Onderzoek | NWO-Vidi 452-12-009 | Christian F Doeller |
| Nederlandse Organisatie voor Wetenschappelijk Onderzoek | NWO-MaGW 406-14-114 | Christian F Doeller |

The funders had no role in study design, data collection and interpretation, or the decision to submit the work for publication.

### Author contributions

LD, Conception and design, Acquisition of data, Analysis and interpretation of data, Drafting or revising the article; JLSB, Conception and design, Acquisition of data, Drafting or revising the article; TNS, Conception and design, Drafting or revising the article; CFD, Conception and design, Analysis and interpretation of data, Drafting or revising the article

### Author ORCIDs

Lorena Deuker, http://orcid.org/0000-0002-4939-5862
Jacob LS Bellmund, http://orcid.org/0000-0002-2098-4487
Tobias Navarro Schröder, http://orcid.org/0000-0001-6498-1846

### Ethics

Human subjects: All participants gave written informed consent, filled in a screening form to ensure they did not meet any exclusion criteria for fMRI and were compensated for their time. The study was approved by the local ethics committee (CMO Regio Arnhem-Nijmegen).

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
