## [Decision Letter]

Thank you for submitting your article "An event map ofmemory space in the hippocampus" for consideration by *eLife*. Your article has been reviewed by two peer reviewers, one of whom, Kenneth A. Norman (Reviewer #2), has agreed to reveal his identity, and the evaluation has been overseen by a Reviewing Editor and Timothy Behrens as the Senior Editor.

The reviewers have discussed the reviews with one another and the Reviewing Editor has drafted this decision to help you prepare a revised submission.

All the reviewers agreed that the research question addresses an important and timely issue regarding how space and time might be represented in the brain. The reviewers also agree that the learning paradigm is clever, especially with respect to the use of teleporters to disentangle actual temporal distance from spatial distance. Indeed, this is themost compelling aspect of the design.

However, the reviewers all agree that new analyses and descriptions of current data need to be included for the paper to be suitable for publication. The relevant issues are presented in more detail below but are summarized here. (1) The data analyses and discussion with respect to whether temporal and spatial memory are independent requires more attention. (2) Analyzing the data based on actual (objective) spatial and temporal distance is required, either way the data work out – as that will help to address whether hippocampus is building an accurate map or only a memorial (less accurate) one. (3) Several other analytic details should be addressed – please see below for details that on the requested data analyses and reporting on this issue of how independent the memory representations are and whether hippocampus also represents objective spatial and temporal distance

Independence of space and time (reviewer comments):

1) The authors suggest in the Discussion that the spatial and temporal dimensions are represented separately. However, it is not clear that they show this, nor am I certain that this is what the authors believe. Although they show that the two dimensions both contribute to neural similarity, this does not mean they are separate representations. For example, they could be additive within a single dimension and still separately contribute to the effect. The results, as presented, appear to be compatible with this "single dimension" idea. If the authors are comfortable with this idea, they can just state this in the paper as a possible explanation, and I would be totally fine with that. If the authors want to show that space and time are truly represented separately in the anterior hippocampus, they would need to show that some regions are significantly more affected by time than space, and vice-versa (currently, they just show that the regions showing a significant time effect are not completely overlapping with the regions showing a significant space effect

2) It's interesting that temporal and spatial distance judgments were correlated in memory, particularly since object pairs were assessed for spatial and temporal distance on separate trials. I don't think such an effect has been shown before and it has interesting implications for overlap in their representation. However, it could also be an inferential bias since spatial and temporal distances tend to be correlated in the real world. Were participants ever told that the two distances were not related? Also, a couple additional analyses might help unpack this data. In particular, you should be able to get a sense for which memory is biasing the other by examining when the direction of error is towards the other distance type (e.g., a spatially close trial judged as further when the temporal distance is longer), or whether the source of the bias is whichever was tested first in the distance test (was this counterbalanced across pairs?)

3) The authors try to remove the shared variance in the MPS analysis. However, I am not sure this will totally solve the issue. Since the two are collinear, I am not sure how you can remove variance from a different task with fMRI and MPS, which is based on the change in object correlations before and after. I guess I would have preferred to see a more detailed consideration behaviorally of exactly how and in what manner the two are correlated vs. independent

4) I think the spatial and temporal combination finding is based on partial overlap of the searchlight clusters. But this doesn't prove that the combined information is stored independently and together. It merely shows the clusters have some overlap functionally, there could be different neurons responding based on different voxel patterns. I think this analysis needs more work-up and consideration. If the clusters overlap, are the correlations themselves the same? The variance in the patterns? Again, more details and consideration is needed here to reach this conclusion

Objective Distance Analysis:

1) One nice feature of this study is the independent manipulation of spatial and temporal (or sequential) distance that could allow for the same analysis to be performed on objective distances rather than mnemonic, or subjective, distances. From a theoretical standpoint, it would be interesting to know which is more predictive of the neural representation. In addition, as the memory measures of distance are correlated, using the objective measures may allow better dissociation between temporal and spatial distance. One challenge in doing this analysis is the absolute temporal distance is not fixed because subjects could move at different speeds. However, since that data is recorded, it should be possible to get the true experimental distance. Alternatively, since the authors designed the study as a full factorial (fully crossing high and low distance in space and time), they could analyze the similarity data in this discretized way rather than using a continuous measure of distance. This addition would help with interpretation of the data and also could add significant novelty, as a dissociation of objective versus subjective distance representations has not been shown before.

Analytic Details:

1) Sphere Size: It is unclear why the authors chose the sphere size they did for the searchlight analysis. It seems that a 9mm radius is quite large for looking within the hippocampus. In particular, the posterior hippocampus can be thin and not spherical in shape. Thus, one possibility is the specificity of the results to the anterior hippocampus could be due to the choice of sphere shape and size. I would recommend iterating through various searchlight sizes to address this concern or adopting the anatomical ROI procedure (splitting the hippocampus into thirds along the long axis) as in the Nielson study. Relatedly, given that the authors have high-resolution data, hippocampal subfield segmentation would a complementary and perhaps more principled way of investigating within hippocampal specificity of distance effects

2) Several details of the analyses were not clear to me and need to be explained in significantly more detail. In Figure 4, it is explained that the correlation coefficients between the RSA change from pre to post navigation, were correlated with the spatial and temporal distance judgments, and then assigned to the center of a search light sphere. I am at a loss in terms of understanding what was done here. If such a correlation exists in the data, the authors really need to show the different distances and RSA values on a scatter plot. It is also unclear to me why some analyses were done for the whole hippocampus ROI and why others were done with a searchlight. Will these methodologies converge? Along these lines, the authors should also show plots with RSA values (ideally, the raw averaged correlations) for near vs. far spatial and temporal distances and their change. It is very difficult to follow exactly how the z values were obtained in the analysis and this needs more explication. Overall, the figures should display more of the raw data (at least, data in an early form) to walk the reader through what was obtained and in what manner. I suggest adding as many new figures to the actual main figures to make these points more transparent and convince the readers of the important findings here.

3) The authors make a strong statement about anterior vs. posterior hippocampus but at least one of the clusters spans both anterior and medial hippocampus (Figure 6B). Given that none of the analyses were done in native space (as far as I can understand, please see Yassa and Stark 2009 Neuroimage for a detailed discussion of this issue in the hippocampus) and the slice thickness was around 1.5, can the authors be confident they are in anterior hippocampus? I don't think the anterior argument adds much to the paper here and the authors might consider deleting it.

Scholarship:

1) In the Introduction, the authors briefly mention two fMRI studies investigating temporal coding in humans. This seems to be an under representation of a literature that has been growing rapidly as indicated by recent reviews of both human and across species work (e.g., Eichenbaum, 2014; Davachi and DuBrow, 2015; Ranganath & Hsieh, 2016). However, the bigger issue is that the main findings of the two highlighted papers are inaccurately described. The Hsieh paper doesn't show only that items closer together have greater pattern similarity (in fact, this would be meaningless due to autocorrelation). Rather they show that hippocampal pattern similarity is sensitive to the conjunction of item identity and temporal position (by showing increased similarity across sequences only for items that share both features). The authors also left out the main (and most relevant) finding of the Ezzyat study, which is that hippocampal pattern similarity tracked subjective temporal distance across boundaries. Without this, it's not clear what the study has to do with temporal coding at all. Also, as this latter study investigated patterns at encoding of trial-unique sequences, the last sentence of that paragraph is not technically correct.

2) A previous paper by Kyle et al. 2015 Behavioral Brain Research showed similar findings for near vs. far temporal distances but the opposite pattern for spatial distances (far RSA > near). This paper should be discussed and overstatements about the novelty here should be avoided in light of this paper although I think differences in the paradigms can probably account for the differences in findings for near vs. far space. Similarly, a recent paper by Copara et al. 2014 J Neurosci showed pattern separation of independent spatial and temporal information during correct memory retrieval following navigation, as is also shown (in part here). This paper should also be properly contextualized.

[Editors' note: further revisions were requested prior to acceptance, as described below.]

Thank you for resubmitting your work entitled "An event map of memory space in the hippocampus" for further consideration at *eLife*. Your revised article has been favorably evaluated by Timothy Behrens (Senior editor), a Reviewing editor, and two reviewers.

The manuscript has been improved but there are some remaining issues that need to be addressed before acceptance, as outlined below

While the revision was very responsive, there remain some further adjustments of the data presentation and tempering of claims that should be addressed before we can proceed with the manuscript. The specific requested changes are appended here

Reviewer comments, edited by the reviewing editor to highlight remaining issues:

1) The single subject pattern similarity plots are overkill and also a bit hard to decipher. If the question the authors are trying to address is about the correspondence between the patterns for ROI (left vs. right hippocampus) and representation type (space vs. time), it might make more sense to plot the correlations between those things across subjects rather than to essentially ask readers to perform that correlation by eye. Having said this, I would be fine with the authors just leaving out these plots entirely.

2) I found the objective distance analysis somewhat unconvincing. Only the space factor leads to significant effects and objective time apparently does not have effects. There were also no additional analyses to really compare with the subjective ratings and given that the two are somewhat correlated anyway, I am not sure what new information is gained here. This analysis also leads to the confusing conclusion that right hippocampus coded remembered distances in both space and time while left hippocampus coded objective and remembered distances in space. This doesn't quite fit with their results in some places (as shown in Figure 5 for the ROI analysis) and has no real precedent in the literature. Laterality findings are also notoriously difficult to replicate with fMRI and overall have not told a coherent story in the literature. I would suggest deleting these speculations and simply focusing on the role of the hippocampus in remembered spatial and temporal distances.

3) I still have concerns about the argument that effects are "localized" to anterior hippocampus. I realize this may fit with some of the arguments from past papers from this lab but it doesn't particularly fit given the data here. The combination of space and time analysis (subsection “Neural changes are modulated strongly by the combination of space and time”) shows a cluster that spans anterior and medial hippocampus. Also, Figure 5—figure supplement 1 clearly shows that the effects are trending in medial and posterior and part of driving the effect overall for the hippocampal ROI. Thus, it is not really correct to focus on anterior here because the authors haven't shown a double dissociation (as they did in their recent Nature Neuroscience paper). It could simply be that the effect is stronger in anterior, perhaps due to features of the scan acquisition. Without an interaction effect (dissociation), subject-specific ROIs (rather than using atlases, as is stated in the Methods), and at least one cluster that spans both anterior and medial, references to anterior specialization should be deleted. Another reason for this is that the majority of spatial effects tend to be in posterior and it is not clear why the spatial distance cluster is in anterior (Figure 5 and Figure 5—figure supplement 1) and does not fit with the literature overall. I therefore suggest much more caution with speculations about anterior/posterior specialization when the methods and data can’t really support this.

4) There continue to be instances of overstatement ("highly" significant: note that an effect either crosses the stated threshold or not). I think another example of this is the argument that this paper somehow tests an issue neglected in past work, which is the connection between episodic memory and navigation. Technically, there is no neural analysis of data during encoding or recollection and thus the paper doesn't exactly address episodic memory specifically, more how representation of spatial metrics change as a function of navigation. While the authors do relate to later memory performance, I think there should be a little more caution in overstating the novelty of their findings. I consider this point though relative minor.

5) Finally, the authors should be careful not to overstate the dissociation between space and time in their paradigm. The subsequent analyses did tease these out although the two variables were in fact correlated in a significant number of subjects. Some restatement of this point in the Discussion is probably warranted.

---

## [Author Response]

All the reviewers agreed that the research question addresses an important and timely issue regarding how space and time might be represented in the brain. The reviewers also agree that the learning paradigm is clever, especially with respect to the use of teleporters to disentangle actual temporal distance from spatial distance. Indeed, this is the most compelling aspect of the design

However, the reviewers all agree that new analyses and descriptions of current data need to be included for the paper to be suitable for publication. The relevant issues are presented in more detail below but are summarized here. (1) The data analyses and discussion with respect to whether temporal and spatial memory are independent requires more attention. (2) Analyzing the data based on actual (objective) spatial and temporal distance is required, either way the data work out – as that will help to address whether hippocampus is building an accurate map or only a memorial (less accurate) one. (3) Several other analytic details should be addressed – please see below for details that on the requested data analyses and reporting on this issue of how independent the memory representations are and whether hippocampus also represents objective spatial and temporal distance

*Independence of space and time (reviewer comments):*

1) The authors suggest in the Discussion that the spatial and temporal dimensions are represented separately. However, it is not clear that they show this, nor am I certain that this is what the authors believe. Although they show that the two dimensions both contribute to neural similarity, this does not mean they are separate representations. For example, they could be additive within a single dimension and still separately contribute to the effect. The results, as presented, appear to be compatible with this "single dimension" idea. If the authors are comfortable with this idea, they can just state this in the paper as a possible explanation, and I would be totally fine with that. If the authors want to show that space and time are truly represented separately in the anterior hippocampus, they would need to show that some regions are significantly more affected by time than space, and vice-versa (currently, they just show that theregions showing a significant time effect are not completely overlapping with the regions showing a significant space effect).

The reviewer raises an important point here about how our results should be interpreted. On the one hand, we have set up our experiment in a way which allows us to investigate memory for spatial and temporal relationships independently, and while the two factors are to come degree correlated in participants’ memory (see new results on this below), we were able to show with our control analyses that both dimensions contribute to the neural similarity increase we observe, as the reviewer also notes. On the other hand, the clusters we find for the two dimensions partially overlap, and we do not directly how that some voxels carry significantly more information about one dimension than about the other. We interpret this finding in such a way that both dimensions are represented by a common memory mechanism which translates relationships between experiences in the real world into an interconnected mental representation of them on a neural level, which we have captured with the term event map. Therefore, we concur with the reviewer that space and time, while making separate contributions are integrated by a common mechanism into one dimension, namely proximity in a multi-dimensional event map. We thank the reviewer for pointing out this lack of clarity in our Discussion. We have made the following changes to the manuscript to take the referee’s suggestion into account:

Results:

“These results suggest that both the dimensions of space and of time contribute to the observed pattern similarity increases.”

Results:

“Thus, part of the overlap in peak regions for space and time may be explained by the two factors being correlated in participants’ memory judgments, but there are also significant effects of space and time after statistically removing the influence of the other factor, suggesting that the two dimensions both contribute to pattern similarity increases in right anterior hippocampus.”

Discussion:

“Our data support the notion of a common hippocampal coding mechanism in space and time: neural similarity scales with the proximity of event memories in both dimensions. […] The observed strong effect for the combination of space and time further suggests that the two dimensions might be integrated in a common dimension in the hippocampus, i.e. spatio-temporal proximity, supporting the formation of hierarchical structures in a memory space (Collin et al., 2015; Eichenbaum et al., 1999; McKenzie et al., 2014).”

*2) It's interesting that temporal and spatial distance judgments were correlated in memory, particularly since object pairs were assessed for spatial and temporal distance on separate trials. I don't think such an effect has been shown before and it has interesting implications for overlap in their representation. However, it could also be an inferential bias since spatial and temporal distances tend to be correlated in the real world. Were participants ever told that the two distances were not related? Also, a couple additional analyses might help unpack this data. In particular, you should be able to get a sense for which memory is biasing the other by examining when the direction of error is towards the other distance type (e.g., a spatially close trial judged as further when the temporal distance is longer), or whether the source of the bias is whichever was tested first in the distance test (was this counterbalanced across pairs?)*

We would like to thank the reviewer for their helpful suggestions with regard to our behavioural data and we have addressed them with a number of new analyses. First of all, we would like to note that participants were indeed explicitly instructed that the spatial and temporal distance ratings were in comes cases quite different from one another. We explicitly mentioned that items could still be close together in space even when they were at different ends of the route (i.e. far apart in time) due to the route leading back and forth through the city, and we also pointed out that items could be far apart in space but close together in time due to the teleporters. Notably, this instruction was only given to participants immediately before the memory judgment task, i.e. after the imaging part was already concluded, so the effect we find in the fMRI data cannot be explained by this explicit instruction. Secondly, we would like to point out that despite significant correlations between the two types of ratings, there was an overall good match between actual distance and remembered distances in most of our participants in both dimensions, which strongly indicates that most participants were treating the two dimensions differentially. Thirdly, with regard to the order in which the two domains were tested, this was not counter-balanced across pairs of items, but the order was random within the two domains. Also, as we tested spatial and temporal memory separately in four blocks each, and one kind of block always had to come first, the order could not be perfectly balanced within a participant (but note that across participants, we randomly assigned whether space or time would be the first block). We report results on the potential impact of order below.

In the course of the new analyses, we have found an error in our code that led to an over-estimation of the extent to which remembered spatial and remembered temporal distance judgments are correlated. The mean +/- std R is 0.31 +/-0.29 instead of 0.40 +/- 0.22, and it is significant in 14 instead of 24 out of 26 participants. We apologize for this error, which we have corrected in our revised manuscript. We still think that the correlation between spatial and temporal distance judgments is too pronounced to be neglected and should be further investigated in the way the reviewers have suggested.

We have tried to unravel the potential biases affecting the distance judgments with the following new analyses (also refer to the new Figure 2):

We estimated the impact of both actual space and actual time on the two memory-based distance ratings by setting up two GLMs, each with two predictors (actual space andactual time) and one criterion, remembered space and remembered time, respectively. We find that there is a significant “cross-over” between the domains, namely that spatial distance explains variance in temporal distance ratings and vice versa. However, the influence of the “cross-over” domain was considerably weaker than that of the “correct” domain (p < 0.0001 for both remembered spatial distance and remembered temporal distance), confirming that, overall, the participants were following the instruction and were producing correct responses. The results of this analysis can be found in the middle panel of a new behavioural results figure (Figure 2). Following up on this and taking up the reviewer’s suggestion, we then explored whether one domain was biased more strongly by the other. For this, we calculated for every pair of items the error in the distance ratings, i.e. the difference in z-scored actual distance and z-scored distance rating. We did this separately for space and time, resulting in “spatial error” values for every pair of items and “temporal error” values for every pair of items. We then took the spatial errors and correlated them with the temporal distance in these trials (both actual and remembered). Across participants, correlation coefficients were significantly different from zero for this error analysis (all p < 0.0001), suggesting that, indeed, the errors were related to what was present in the other domain. However, the correlation coefficients were not significantly different from one another across participants, suggesting that no domain was biasing the other more than vice versa. These results can be found in the lower left panel of the new behavioural figure (Figure 2).

To assess the potential impact of test order, we repeated the error analysis described above but this time differentiated depending on the order of recall: We split up those trials in which the spatial domain was probed first and those trials in which the temporal domain was probed first during memory recall. If there was a relationship with test order, then spatial distance should have a higher impacton errors in temporal ratings when space was tested first, and the other way around. However, there were no differences in the strength of bias in the different order conditions. These results can be found in the lower right panel of the new behavioural figure (Figure 2).

We have summed up the results of these new analyses in a new figure and replaced the current Figure 2 with this new figure. The original Figure 2 contains participant-specific scatter plots but much less global information and we thus moved Figure 2 to the supplement instead (Figure 2—figure supplement 1).

Also, in the course of the new analyses we have improved consistency in our behavioural measures. Whereas in Figure 1—figure supplement 2, we show the median walking time from one object to the next, we had so far calculated actual temporal distance between objects by using mean walking time. We have updated this now so that actual temporal distance is always based on median walking time across repetitions of the route, as this is a more robust metric of objective temporal distance. As can be seen in the updated participant-specific scatter plots, this new measure does not change the overall result pattern (only subtle changes in mean R for remembered temporal distances, which is 0.64 instead of 0.65, and a slightly smaller T-value for the difference between temporal accuracy and spatial accuracy (T_25_ = 2.52 instead of T_25_ = 2.57) and corresponding slightly higher p value (0.019 instead of 0.0166)).

In sum, these new results show no evidence for a differential influence of one of the dimensions in participants’ memory. The revised sections of the manuscript read as follows:

Results:

“For temporal judgments, memory distances were significantly correlated with actual temporal distances in 24 of the 26 participants (p < 0.05; R = 0.64 ± 0.29 (mean ± std), see Figure 2A for the correlation coefficients across participants and Figure2-figure supplement—figure supplement 1 for participant-specific scatterplots). […] While spatial and temporal distance ratings were correlated with one another in some participants, there is no evidence that either the spatial domain or the temporal domain had a bigger impact on distance ratings than the other, and that the degree to which one domain was biased by the other did not depend on which domain was tested first.”

Materials and methods:

“Participants were explicitly instructed that the spatial and temporal distance ratings were in some cases quite different from one another. […] Notably, this instruction was only given to participants immediately before the distance judgment task, i.e. after the imaging part was already concluded, so the effects we find in the fMRI data cannot be explained by this explicit instruction.”

We have included a new figure which depicts results from the behavioral analyses and replaces the previous Figure 2, which is now Figure 2—figure supplement 1.

*3) The authors try to remove the shared variance in the MPS analysis. However, I am not sure this will totally solve the issue. Since the two are collinear, I am not sure how you can remove variance from a different task with fMRI and MPS, which is based on the change in object correlations before and after. I guess I would have preferred to see a more detailed consideration behaviorally of exactly how and in what manner the two are correlated vs. independent.*

We believe that this comment is in part related to the previous point. We have added several new analyses to disentangle spatial and temporal distance ratings. To summarize these new results: Firstly, the ratings for the two domains are correlated, yet they are also mostly accurate for each domain. Secondly, when testing the influence of the actual distance in the two domains on spatial and temporal distance ratings directly with two separate GLMs, we find that actual distances in space and time explain distance ratings in both space and time, i.e. there is a “cross-over” between dimensions. But spatial ratings are explained better by actual spatial distance, and temporal ratings are explained better by actual temporal distance. Thirdly, errors in one domain, i.e. discrepancies between actual distance and distance rating, are biased towards the distance in the other domain, but no domain is more strongly affected than the other. In sum, this leads us to believe that the ratings for each domain are most strongly related to the actual distance in that domain, but that the distance in the other domain also has an impact, even if it is much smaller. Still, when we observe a relationship between distance ratings in one domain and pattern similarity increase on a neural level as we do, it could be that all or part of the effectis driven by a third factor, i.e. participant’s memory of the other domain. By regressing out this third factor (e.g. spatial distance ratings) from the pattern similarity increases before we correlate the factor of interest (e.g. temporal distance ratings) with the residuals of that GLM, we believe that we are getting a cleaner estimate of the relationship between each behavioural factor of interest and the neural data. To make the rationale for this approach more clear, we have extended the explanation of the procedure in both Methods and Results.

Results:

“However, behavioral analyses had revealed that some participants’ spatial and temporal distance judgments were correlated, even though spatial and temporal distances between items were designed to be independent in the task. […] Therefore, we investigated in an additional analysis whether there were separate contributions of the two factors: First, we removed variance explained by spatial distance judgments from the pattern similarity changes in pairs of items in a GLM, and correlated the residuals from this model (i.e. what could not be explained by spatial distance judgments) with temporal distance judgments.”

Materials and methods:

“Because temporal and spatial distance estimates were correlated in some participants’behavioral ratings, it could be that effects we find for one domain can be explained by the correlation with the other domain, and that the domain has no unique contribution to the effect we find. […] Then, we took the residuals of this GLM (i.e. variance not explained by remembered spatial distance) and correlated these residuals with the remembered temporal distances.”

*4) I think the spatial and temporal combination finding is based on partial overlap of the searchlight clusters. But this doesn't prove that the combined information is stored independently and together. It merely shows the clusters have some overlap functionally, there could be different neurons responding based on different voxel patterns. I think this analysis needs more work-up and consideration. If the clusters overlap, are the correlations themselves the same? The variance in the patterns? Again, more details and consideration is needed here to reach this conclusion.*

The reviewer raises an important point. We agree that our manuscript benefits from providing more of the underlying ‘raw’ data (whichis also related to another point about clarity of the description of procedures below). Therefore, we have included as supplemental figures the similarity increase matrices for the three ROIs (bilateral hippocampus, left hippocampus,right hippocampus) for every participant, as well as for the respective peak voxels of our three analyses in the searchlight approach (space, time, combination). We believe that by looking at these raw data, it becomes clearthat the pattern similarity increase matrices are not necessarily very similar within a participant in the three different ROIs or in the respective peakvoxels of the three different analyses, even though the peak voxels are quite close together, as the reviewers point out. This makes sense when considering that even in the case of overlap between ROIs or the searchlights, many voxels will still be different – and as the pattern similarity is calculated over the pattern of all voxels within a ROI or searchlight, changing even very few voxels might have considerable impact on the overall pattern similarity. We included 4 new figures with pattern similarity increase matrices as figure supplements. We think these additional figures will be very useful for readers both in terms of following the analysis steps and in assessing the pattern similarity increases underlying our main effects. In addition to the new figures, we have changed wording with regard to time and space being stored independently and together, also in response to previous points (see responses to point 1 and 2 above).

Results:

“These results suggest that both the dimensions of space and of time contribute to the observed pattern similarity increases.”

Results:

“Thus, part of the overlap in peak regions for space and time may be explained by the two factors being correlated in participants’ memory judgments, but there are also significant effects of space and time after statistically removing the influence of the other factor, suggesting that the two dimensions both contribute to pattern similarity increases in right anterior hippocampus.”

Discussion:

“Our data support the notion of a common hippocampal coding mechanism in space and time: neural similarity scales with the proximity of event memories in both dimensions. […] The observed strong effect for the combination of space andtime further suggests that the two dimensions might be integrated in a common dimension in the hippocampus, i.e. spatio-temporal proximity, supporting the formation of hierarchical structures in a memory space (Collin et al., 2015; Eichenbaum et al., 1999; McKenzie et al., 2014).”

We included two supplemental figures which depict pattern similarity increase matrices forevery participant in the three ROIs (Figure 4—figure supplement 2; bilateral hippocampus, left hippocampus, right hippocampus) and for the three respective peak voxels of our three main searchlight analyses (Figure 4—figure supplement 3; remembered spatial distances, remembered temporal distances and the combination of both). For display quality, (A) shows the first half of participants, (B) shows the other half of participants.

*Objective Distance Analysis:*

*1) One nice feature of this study is the independent manipulation of spatial and temporal (or sequential) distance that could allow for the same analysis to be performed on objective distances rather than mnemonic, or subjective, distances. From a theoretical standpoint, it would be interesting to know which is more predictive of the neural representation. In addition, as the memory measures of distance are correlated, using the objective measures may allow better dissociation between temporal and spatial distance. One challenge in doing this analysis is the absolute temporal distance is not fixed because subjects could move at different speeds. However, since that data is recorded, it should be possible to get the true experimental distance. Alternatively, since the authors designed the study as a full factorial (fully crossing high and low distance in space and time), they could analyze the similarity data in this discretized way rather than using a continuous measure of distance. This addition would help with interpretation ofthe data and also could add significant novelty, as a dissociation of objectiveversus subjective distance representations has not been shown before.*

We have taken up the reviewer’s helpful suggestion and performed the same analyses on the objective distances that we have performed for the subjective (remembered) distances. We defined actual spatial distance as the Euclidean distance between pairs of items, and we defined actual temporal distance between pairs of itemsas the median walking time from one item to the other across the different repetitions of the route during the navigation task. It should be noted that in some cases participants got lost on the route from one object to another, leading to unusually high walking time in one repetition as compared to others; however, the median should be robust against outliers like these. Analogous to our ROI approach for subjective distances, we correlated actual spatial distance, actual temporal distance and a combination of both with pattern similarity increases across all hippocampal gray-matter voxels (and right and left hippocampus separately). In summary, this ROI analysis revealed that actual spatial distance was correlated with pattern similarity increases across gray-matter voxels in bilateral hippocampus. Interestingly, temporal distances were not reflected in pattern similarity increases in either bilateral, right or left hippocampus, and neither was the combination of objective spatial and objective temporal distances.

As the reviewer also points out, unlike the behavioural/subjective distance ratings, the objective spatial and temporal distances are unrelated to one another and can be split up into two factors (spatial distance versus temporal distance) with two steps each (high and low distance). To follow up on this interesting suggestion, we performed another additional analysis, implementing the ANOVA suggested by the reviewers. For this, we assigned every pair of items to one offour fields (space high and time high, space high and time low, space low andtime high, space low and time low), and averaged pattern similarity increasesin our three ROIs (bilateral hippocampus, left and right hippocampus) for pairs of items belonging to these four fields separately. We then tested with 2-wayrepeated measures ANOVA whether there was a significant effect across participants for the two factors and their interaction. We find a significant effect for the factor ‘space’ in bilateral hippocampus (F_1,25_ = 8.29,p = 0.008) and in left hippocampus (F_1,25_ = 8.84, p = 0.006), but not in right hippocampus (p > 0.05). The factor ‘time’ was not significant in any of the three ROIs, and neither was the interaction between the two factors (all p > 0.05).

These results are interesting because they suggest that there might be a different pattern of neural representation for objective vs. subjective/remembered distances, which to our knowledge has not been shown before. It seems that right hippocampus, rather than storing an objective map of spatio-temporal distances, supports the representation of subjective spatial and temporal distances, a function which is more closely related to what we have described as an event-map, whereas left hippocampus carries information on subjective and objective distances, but for the spatial domain only. We think this finding might lead to a more thorough investigation of the relationship between objective and subjectives patio-temporal distances and their neural representation in future studies. Therefore, we thank the reviewers for this suggestion and describe the results of this new analysis in the revised manuscript and discuss possible implications.

Results:

“Impact of objective spatial and temporal distance on pattern similarity:

So far, we investigated how the remembered spatial and temporal relationships between items are reflected in pattern similarity increases in the brain. […] Right hippocampus – rather than storing an objective map of spatio-temporal distances– supports the representation of subjective spatial and temporal distances, a function which is more closely related to what we have described as anevent-map, whereas left hippocampus carries information on both subjective andobjective distances, but for the spatial domain only.

Discussion:

“One interesting finding here is that we observed a different pattern of results for neural representation of objective spatial distances compared to remembered distances. […] It is very likely that other factors in addition to objective spatial and temporal distance impact how a spatio-temporal event map is constructed and remembered, and it will be very interesting in future studiesto identify these factors.”

*Analytic Details:*

*1) Sphere Size: It is unclear why the authors chose the sphere size they did for the searchlight analysis. It seems that a 9mm radius is quite large for looking within the hippocampus. In particular,the posterior hippocampus can be thin and not spherical in shape. Thus, one possibility is the specificity of the results to the anterior hippocampus couldbe due to the choice of sphere shape and size. I would recommend iterating through various searchlight sizes to address this concern or adopting the anatomical ROI procedure (splitting the hippocampus into thirds along the longaxis) as in the Nielson study. Relatedly, given that the authors have high-resolution data, hippocampal subfield segmentation would a complementary and perhaps more principled way of investigating within hippocampal specificity of distance effects.*

We think that incomparison to other fMRI studies using a searchlight approach, our sphere size of 9mm radius around the center voxel is not unusually large (compared for example with 10mm in Chadwick et al., 2015). In addition, to estimate pattern similarity, we have to include a certain minimum number of voxels (set to 30 in our study), so reducing sphere size by much will interfere with this restriction, especially as we only include gray-matter voxels within a sphere.

However, we do acknowledge that the reviewer makes a valid point and that with our primary focus on the hippocampus, 9mm radius sphere size might lead to missing some areas that show a significant effect, especially where the hippocampus is thin. Therefore, we have set up two additional analyses. Firstly, we implemented the ROI approach suggested by the reviewer, splitting hippocampus into thirds along the longitudinal axis with an identical protocol as previously used in our group (cf. Collin et al., 2015; posterior portion of the hippocampus: from Y =-40 to -30; mid-portion of the hippocampus: from Y = -29 to -19; anterior portion of the hippocampus: from Y = -18 to -4). We included a figure of the ROI and results (Figure 5—figure supplement 1). In short, we find that only anterior hippocampus is significantly related with remembered spatial andtemporal distances and the combination of them, consistent with what we find in the searchlight approach.

Secondly, we also re-ran our searchlight analysis with three additional sphere sizes (6mm, 7.5mmand 10.5mm). We have included a figure below which shows that we find more or less the same significant clusters in right hippocampus across the different searchlight sizes (with exception of the cluster for remembered spatial distances at the smallest searchlight size, which does not survive small volume correction for bilateral hippocampus). Even though it is reassuring that our effects are preserved across different sphere sizes, we think including this analysis would not add much information to our manuscript and would therefore prefer not to include it in the manuscript unless the reviewers consider it as essential for the reader.

The reviewer’s suggestion to perform subfield analysis is certainly a very interesting one. However, as we did not have any specific hypotheses about subfield contributions, we did not specifically design our experiment in a way thatwould be optimal for subfield analysis. For example, we did not collect a T_2_ weighted structural image, which would have been crucial for reliable delineation of subfields. Therefore, we do not think this is a feasibleanalysis for our data.

We have included a supplemental figure which shows the results of an additional ROI analysis in which hippocampus was split in thirds along the hippocampal long axis (Figure 5—figure supplement 1).

Author response image 1 shows the results for our main effects when using different searchlight spheres in our analysis, which we suggest not to include in the manuscript.

Author response image 1.Effects are preserved across different searchlight sizes.The effects of the three main analyses are similar when the radius of the searchlight sphere is smaller (6mm and 7.5mm) or bigger (10.5mm) than the original radius (9mm) – with the exception of the remembered spatial distance effect at the smallest sphere size, which does not survive small volume correction for bilateral hippocampus. All clusters shown here survive small volume correction for bilateral hippocampus and are located in the right hemisphere. Note that in order to get a good estimate of pattern similarity across voxels, a minimum number of graymatter voxels should be included in the searchlight (set to 30 in this study). At the smallest sphere size, this restriction is not met for aconsiderable number of voxels, and therefore, no values are calculated forthese searchlights. This is illustrated by showing the masks in the right column, which are the masks of all voxels for which all participants have values for the respective sphere size.**DOI:**
http://dx.doi.org/10.7554/eLife.16534.016

*2) Several details of the analyses were not clear to me and need to be explained in significantly more detail. In Figure 4, it is explained that the correlation coefficients between the RSA change from pre to post navigation, were correlated with the spatial and temporal distance judgments, and then assigned to the center of a search light sphere. I am at a loss in terms of understanding what was done here. If such a correlation exists in the data, the authors really need to show the different distances and RSA values on a scatter plot. It is also unclear to me why some analyses were donefor the whole hippocampus ROI and why others were done with a searchlight. Will these methodologies converge? Along these lines, the authors should also show plots with RSA values (ideally, the raw averaged correlations) for near vs. far spatial and temporal distances and their change. It is very difficult to follow exactly how the z values were obtained in the analysis and this needs more explication. Overall, the figures should display more of the raw data (atleast, data in an early form) to walk the reader through what was obtained and in what manner. I suggest adding as many new figures to the actual main figures to make these points more transparent and convince the readers of the important findings here.*

We would like to thank the reviewers for pointing out how our descriptions of methodological procedures and data can be improved. We carefully went through the descriptions and believe that the main reason for lack of clarity is that we have depicted only the searchlight analysis in greater detail. We have added a new figure, which demonstrates the procedure for the ROI analysis, both on first-level (single subject) and on second-level (across subjects) analyses. We have also adapted the previous searchlight methods figure as well as the manuscript tomake clear that the two approaches (ROI and searchlight) are mostly analogous, but provide complementary information: the ROI analysis for testing a strong a-priori hypothesis (i.e. that hippocampus will be involved in therepresentation of spatio-temporal distances), and the searchlight analysis for a) potentially finding are as outside of hippocampus that show the same effect and b) pinpointing the effect more locally within the hippocampus.

In both ROI and searchlightfigures, we have included example data plots to better illustrate the individual steps that were taken and have linked them to figure supplements with corresponding raw data in all subjects where appropriate. One difficulty in visualizing the analysis is that much of the crucial steps occur on the single subject level, where RSA data is inherently noisy and it is difficult to extract regularities visually; the robustness of our findings arises in second-level analyses, when consistency across the single-subject data can be discerned. For the searchlight approach, another difficulty is that all plotting of raw data could potentially be done for each of the tens of thousands of center voxels. We have therefore decided to plot raw data for the respective peak voxels of our three main analyses only. We feel that by addingmore of the raw data, as the reviewers suggested, our paper is improved both interms of clarity and transparency.

Following another reviewer suggestion, we have revised Figure 5 and included a plot with raw pattern similarity increases, averaged according to whether remembered distances were low versus high (dividing all pairs of items with a median split). We show bars for our three ROIs (bilateral, left and right hippocampus) in our three conditions (remembered spatial distance, remembered temporaldistance, and combination of both). We think that showing the data underlying our correlation analysis (shown in panels B and C of the same figure) in this way makes them more accessible to the reader. Also, this displaying of the datais now more consistent with Figure 6, depicting the results of the searchlight analysis, in which we also show the averaged raw pattern similarity increases. We have made the following changes in the manuscript:

Throughout manuscript:

When werefer to the change in pattern similarity from the picture viewing task pre (PVT pre) to the picture viewing task post (PVT post), we had so far described this as R’. However, as we usually write about pattern similarity (PS), wethink the acronym PS’ is more intuitive to capture that we are talking about the difference in pattern similarity from PVT pre to PVT post. We have adapted this throughout the manuscript.

Results:

“Therefore, we related the difference in pattern similarity from PVT pre to PVT post (PS’) to the remembered temporal and spatial distances, both in a region of interest (ROI) analysis and a searchlight analysis (see Figure 4 and Materials and methodsfor details on analyses and nonparametric statistical procedures). We pursued these approaches in parallel because they offer complementary advantages: the ROI approach allows forrigorous testing of a clear a priori hypothesis, while the searchlight approachallows us to identify possible regions outside of hippocampus that show thesame effect, as well as to pinpoint any effect more locally within hippocampus.”

Materialsand methods:

“Subtracting the PVT pre similarity matrix from the PVT post matrix resulted in a matrix that reflected the change in pattern similarity for all pairwise comparisons of items that is due to the learning task (see Figure 4—figure supplement 2 and Figure4—figure supplement 3for the difference matrices in the three areas and in specific voxels from thesearchlight analysis, respectively). […] Thus, for every participant, we gained a surrogate distribution of Spearman correlation coefficients, which were based on shuffled data and compared our real correlation coefficient for a given ROI against this random distribution (see Figure 4 for an illustration of this procedure).”

We adaptedthe previous Figure 4, depicting the analysis procedure for the searchlight approach, to more closely resemble the ROI methods figure (new Figure 4) and moved it to the supplement as Figure 4—figure supplement 1.

We revised Figure 5, in which we have included averaged pattern similarity increases for pairs of items with low vs. high distances between them.

*3) The authors make a strong statement about anterior vs. posterior hippocampus but at least one of the clusters spans both anterior and medial hippocampus (Figure 6B). Given that none of the analyses were done in native space (as far as I can understand, please see Yassa andStark 2009 Neuro image for a detailed discussion of this issue in the hippocampus) and the slice thickness was around 1.5, can the authors be confident they are in anterior hippocampus? I don't think the anterior argument adds much to the paper here and the authors might consider deleting it.*

We agree that the distinction between anterior hippocampus and medial hippocampus might not be very clear from our searchlight results. Even though we performed our analysis on functional data, the two functional scan sessions for the picture viewingtasks (pre and post) had been coregistered to a participant-specific wholebrain image and thus effectively introducing a spatial smoothing of the data. Also, for second level testing, the participant specific brain maps were transformed to the MNI template to enable across-participant comparisons. We have adapted the Discussion to take this point into account.

Results:

“We found a highly significant effect in right medial to anterior hippocampus (see Figure 6B, peak MNI: 32/-17/-22, T25 = 6.07; pcorr < 0.0001, small volume correction).”

Discussion:

“However, no cluster survived correction for multiple comparisons outside of hippocampus and the observed effects in the searchlight analysis were very specific to a region that is located between medial and anterior hippocampus (a clear distinction is difficult to draw due to the coregistration to MNI space). Interestingly, anterior hippocampus has recently been suggested to contain large-scale representations in a memory hierarchy (Collin et al., 2015; cf. McKenzie et al., 2014), which might correspond to the finding that the ventral hippocampus in rats represents the global event context (Komorowski et al., 2013).”

Figure legend Figure 6B:

“The effect was strongest when the two factors of space and time were combined and spans the border between medial and anterior hippocampus.”

*Scholarship:*

*1) In the Introduction, the authors briefly mention two fMRI studies investigating temporal coding in humans. This seems to be an under representation of a literature that has been growing rapidly as indicated by recent reviews of both human and across species work (e.g.,Eichenbaum, 2014; Davachi and DuBrow, 2015; Ranganath & Hsieh, 2016). However, the bigger issue is that the main findings of the two highlighted papers are inaccurately described. The Hsieh paper doesn't show only that items closer together have greater pattern similarity (in fact, this would be meaningless due to autocorrelation). Rather they show that hippocampal pattern similarity is sensitive to the conjunction of item identity and temporal position (by showing increased similarity across sequences only for items that share bothfeatures). The authors also left out the main (and most relevant) finding of the Ezzyat study, which is that hippocampal pattern similarity tracked subjective temporal distance across boundaries. Without this, it's not clear what the study has to do with temporal coding at all. Also, as this latter study investigated patterns at encoding of trial-unique sequences, the last sentence of that paragraph is not technically correct.*

We agree with there viewer that we should have devoted more of the Introduction to the rapidly growing field of temporal coding in the hippocampus. We have added the suggested references and have extended our description of the studies by Hsieh et al. and Ezzyat et al. to more accurately capture their relevant findings.

Introduction:

These findings have led to a re-examination of the hippocampus’ role in temporal memory in rodents and humans (Eichenbaum, 2014; DuBrow and Davachi, 2015; Hsiehand Ranganath, 2016) and to several recent neuroimaging studies in humans. […]More importantly, none of the studies mentioned above compared changes in neural pattern similarity from before the acquisition of the spatial and temporal structure to after.”

We added the following references:

Davachi, L.,DuBrow, S., (2015). How the hippocampus preserves order: the role of predictionand context. Trends in Cognitive Sciences 19, 92–99.

Eichenbaum,H., (2014). Time cells in the hippocampus: a new dimension for mappingmemories. Nat. Rev. Neurosci. 15, 732–744.

Ranganath,C., Hsieh, L., (2016). The hippocampus: a special place for time. Ann. Ny.Acad. Sci. 1369, 93–110.

*2) A previous paper by Kyle et al. 2015 Behavioral Brain Research showed similar findings for near vs. far temporal distances but the opposite pattern for spatial distances (far RSA >near). This paper should be discussed and overstatements about the novelty here should be avoided in light of this paper although I think differences in the paradigms can probably account for the differences in findings for near vs. far space. Similarly, a recent paper by Copara et al. 2014 J Neurosci showed pattern separation of independent spatial and temporal information duringcorrect memory retrieval following navigation, as is also shown (in part here). This paper should also be properly contextualized.*

We thank the referees for pointing us to the studies by Kyle et al. and Copara etal. We have now added the two references to our Discussion and relate them to the findings of our study.

Discussion:

“In humans, it has also been shown that hippocampal damage leads to impairments in both spatial and temporal memory tasks (Spiers et al., 2001; Konkel et al., 2008)and that the hippocampus is active during active retrieval of temporalsequences as well as spatial layouts (Ekstrom et al., 2011), even though dissociable networks for the two retrieval domains were observed outside of the hippocampus. […] Notably, this effect is still present for each domain after statistically controlling for the effect of the other domain, suggesting that both space and time contribute to the observed pattern similarity increase, possibly in an additive manner.”

We haveadded the following references:

Copara, M.,Hassan, A., Kyle, C., Libby, L., Ranganath, C., Ekstrom, A., (2014).Complementary roles of human hippocampal subregions during retrieval of spatiotemporal context. J. Neurosci. 34, 6834–42.

Kyle, C., Smuda, D., Hassan, A., Ekstrom, A., (2015). Rolesof human hippocampal subfields in retrieval of spatial and temporal context.Behav Brain Res 278, 549–558.

[Editors' note: furtherrevisions were requested prior to acceptance, as described below.]

*The manuscript has been improved but there are some remaining issues that need to be addressed before acceptance, as outlined below: While the revision was very responsive, there remain some further adjustments of the data presentation and tempering ofclaims that should be addressed before we can proceed with the manuscript. The specific requested changes are appended here: Reviewer comments, edited by the reviewing editor to highlight remaining issues: 1) The single subject pattern similarity plots are overkill and also a bit hard to decipher. If the question the authors are trying to address is about the correspondence between the patterns for ROI (left vs. right hippocampus) and representation type (space vs. time), it might make more sense to plot the correlations between those things across subjects rather than to essentially ask readers to perform that correlation by eye. Having said this, I would be fine with the authors just leaving out these plots entirely.*

We agree with the reviewers that the single subject pattern similarity plots might be difficult to assess for the reader and that they might not add substantial value to the manuscript. We have therefore removed the supplemental figures containing the plots as well as any reference to them from the manuscript.

*2) I found the objective distance analysis somewhat unconvincing. Only the space factor leads to significant effects and objective time apparently does not have effects. There were also no additional analyses to really compare with the subjective ratings and given that the two are somewhat correlated anyway, I am not sure what new information is gained here. This analysis also leads to the confusing conclusion that right hippocampus coded remembered distances in both space and time while left hippocampus coded objective and remembered distances in space. This doesn't quite fit with their results in some places (as shown in Figure 5 for the ROI analysis) and has no real precedent in the literature. Laterality findings arealso notoriously difficult to replicate with fMRI and overall have not told a coherent story in the literature. I would suggest deleting these speculations and simply focusing on the role of the hippocampus in remembered spatial and temporal distances.*

We agree with the reviewers that interpretation of the results from the objective distance analysis especially with regard to laterality might be premature at this point, and we also feel that the focus of the Discussion should be on the results from the remembered spatial and temporal distance analysis, as this more closely reflects the idea of an event map. For the sake of completeness, we suggest that we still report the results on the objective distance analysis, unless the reviewers or editors feel it would be better to completely leave these data out. However, in accordance with the reviewers’ suggestion, we have removed speculations from the Discussion as to the interpretation of these results.

The passages about the objective distance analysis now read as follows:

Results:

“These results suggest that there might be a different pattern of results for objective spatial and temporal distances as compared to remembered spatial and temporal distances. While we think that the remembered distances more accurately reflect the notion of an event map, it would certainly be very interesting to investigate possible differences in the representation of objective distances in future studies, maybe by systematically increasing divergence between objective distances and remembered distances through experimental manipulation.”

Discussion:

“One interesting finding here is that we observed a different pattern of results regarding the neural representation of objective spatial distances compared to remembered distances. It is conceivable that the spatial and temporal distancesas they are remembered are more indicative of the event map which participants have formed, but the different pattern of results for the objective distances raises the interesting question how objective distances are translated into subjectively remembered distances, and how this is reflected in the neural representation. In our behavioral analyses we found that memory judgments in one domain were biased by the distances in the other domain, but no domain seemed to have a higher impact than the other. It is very likely that other factors in addition to objective spatial and temporal distance impact how a spatio-temporal event map is constructed and remembered, and it will be very interesting in future studies to identify these factors.”

3) I still have concerns about the argument that effects are "localized" to anterior hippocampus. I realize this may fit with some of the arguments from past papers from this lab but it doesn't particularly fit given the data here. The combinationof space and time analysis (subsection “Neural changes are modulatedstrongly by the combination of space and time”) shows a cluster that spans anterior and medial hippocampus. Also, Figure 5—figure supplement 1 clearly shows that the effects are trending in medial andposterior and part of driving the effect overall for the hippocampal ROI. Thus,it is not really correct to focus on anterior here because the authors haven't shown a double dissociation (as they did in their recent Nature Neurosciencepaper). It could simply be that the effect is stronger in anterior, perhaps due to features of the scan acquisition. Without an interaction effect (dissociation), subject-specific ROIs (rather than using atlases, as is stated in the Methods), and at least one cluster that spans both anterior and medial, references to anterior specialization should be deleted. Another reason for this is that the majority of spatial effects tend to be in posterior and it is not clear why the spatial distance cluster is in anterior (Figure 5 and Figure 5—figure supplement 1) and does not fit with the literature overall. Itherefore suggest much more caution with speculations about anterior/posteriorspecialization when the methods and data can’t really support this.

We agree with the referee that localization to anterior hippocampus is not supported by a double dissociation as with previous studies from our lab, and in our main ROI analysis we show that the entire hippocampus supports the pattern similarity increase. In addition, potential differences between anterior and posterior hippocampus are not crucial for the interpretation of our results. We have therefore deleted references to a specialization of anterior hippocampus throughout the manuscript. More specifically, when we report the results of the searchlight analysis, we have changed wording from “right anterior hippocampus”to “right medial to anterior hippocampus”. In addition, we have adapted a paragraph in the Discussion by removing comparisons to other studies which report significant findings in anterior hippocampus, because these comparisons might be too speculative.

We have changed the subtitle for the searchlight Results section, so it now reads as follows: “Searchlight analysis of event map-related neural pattern similarity changes”.

Results:

“But are there regions in hippocampus that more involved in this effect, and are there any other brain regions that show the same pattern?”

Results:

“Taken together, these results show that temporal relationships between events in episodic memory are reflected in pattern similarity changes in a cluster in right hippocampus extending from the medial to the anterior part.”

Results:

“The only observed cluster was thus again located in medial to anterior right hippocampus.”

Discussion: (we have removed comparisons to studies which report effects in anterior hippocampus):

“However, no cluster survived correction for multiple comparisons outside of hippocampus. Therefore, we believe that our findings reflect memory for spatial and temporal relationships, rather than visual similarity.”

4) There continue to be instances of overstatement ("highly" significant: note that an effect either crosses the stated threshold or not). I think another example of this is the argument that this paper some how tests an issue neglected in past work, which is the connection between episodic memory and navigation. Technically, there is no neural analysis of data during encoding or recollection and thus the paper doesn't exactly address episodic memory specifically, more how representation of spatial metrics change as a function of navigation. While the authors do relate to later memory performance, I think there should be a little more caution in overstating the novelty of their findings. I consider this point though relative minor.

We apologize for these over statements and changed the two issues according to the reviewers’ suggestions. In addition, we carefully went through the manuscript and adapted the wording in cases where we felt it could also be perceived as too strong.

Introduction:

“However, it remains elusive how inter-event relationships along multiple dimensions, such as *space* and *time*, are combined and converted into a multi-dimensional mnemonic event map, which might potentially support episodic memory.”

Introduction:

“The purpose of this task was to provide a learning experience for participants in which 16 objects were arranged consistently in a spatial and temporal structure, defined through the complex network of inter-object *relations*.”

Results:

“We found a significant effect in right medial to anterior hippocampus (see Figure 6B, peak MNI: 32/-17/-22, T25 = 6.07; pcorr < 0.0001, small volume correction).”

Discussion:

“Here, we take a first step towards demonstrating such a common coding mechanism in the human hippocampus by showing that both spatial and temporal relationships between events might be represented by a similar mechanism.”

Discussion:

“Secondly, we directly relate the specific neural changes we observe to the interrelations of the memories that have been formed.”

Discussion:

“By showing that both the temporal and the spatial relationships between multiple events are represented in the hippocampus, we took a first step towards unraveling the link between the multi-faceted external world, participants’ memories of it and the neural coding mechanisms supporting the formation of a multi-dimensional mnemonic structure.”

*5) Finally, the authors should be careful not to overstate the dissociation between space and time in their paradigm. The subsequent analyses did tease these out although the two variables were in fact correlated in a significant number of subjects. Some restatement of this point in the Discussion is probably warranted.*

We agree with the reviewers that the dissociation between space and time should not be emphasized too much given our behavioural findings and that we should mention it more explicitly in the Discussion. We have adapted our wording in the Discussion and in the Figure 1 legend so that it does not suggest independence between the factors. In the Methods section, when we describe that in the design of the task we set up the two factors to be independent, we add a caveat that in participants’ memory, the factors were sometimes correlated. We have also added references to these behavioural results in the Discussion.

We have made the following changes to the manuscript:

Discussion:

“Thirdly, we combine spatial and temporal aspects in our learning task and use teleporters to reduce overlap between spatial and temporal distances. […] However, the generally good fit between responses and actual distances and results from the additional analyses in which we statistically control for the influence of the other factor indicate that participants were able to represent the two dimensions separately, at least to a certain degree.”

Discussion:

“Notably, while we found that spatial and temporal distance are to some degree correlated in participants’ memory, the observed effect is still present for each domain after statistically controlling for the effect of the other domain, suggesting that both space and time contribute to the observed pattern similarity increase, possibly in an additive manner.”

Materials and methods section:

“This was necessary for rendering the two factors of time and space independent of each other (see Figure 1—figure supplement 3 for a comparison of spatial and temporal distances; also note that in some participants’ memory, the two factors were not independent from one another, but correlated).”

Figure 1, figure legend:

“Crucially, the spatial and temporal distance between objects was systematically manipulated (see Materials and methods for details).”